# Kernel-based Maximum-of-difference Test for Two-sample Comparison

**Dan Pu** [1 2]  **Tianyi Zhu** [1]  **Yao Yan** [1 3]  **Wei Lan** [3 4]

## Abstract

Two-sample comparison is a fundamental problem in machine learning, with broad applications such as generative modeling. Although the maximum mean discrepancy (MMD) is widely used, MMD-based tests often exhibit poor or even counterintuitive performance under covariance- and location-shift alternatives, partly due to cancellation effects induced by their sum-of-differences construction. To address this issue, we propose a kernel-based maximum-of-difference (MOD) test, which maximizes the squared discrepancy between within-sample and between-sample average distances, thereby improving sensitivity to subtle distributional differences. We further develop a fused MOD procedure to adaptively combine multiple kernels. Extensive experiments demonstrate clear performance gains over existing MMD-based methods.

## 1. Introduction

Testing whether two underlying distributions are identical is a fundamental problem in modern data science. Specifically, given two independent samples $X_1^{(1)}, \ldots, X_{n_1}^{(1)} \overset{\text{i.i.d.}}{\sim} P_1$ and $X_1^{(2)}, \ldots, X_{n_2}^{(2)} \overset{\text{i.i.d.}}{\sim} P_2$, where $P_1$ and $P_2$ are probability distributions on $\mathbb{R}^p$ and $n = n_1 + n_2$ denotes the total sample size, we aim to test the null hypothesis

$$H_0 : P_1 = P_2 \quad \text{against} \quad H_1 : P_1 \neq P_2.$$

This classical two-sample testing problem arises broadly across the scientific literature. For instance, in gene set analysis, one is often interested in assessing whether a gene is differentially expressed under treatment and control conditions (Fox & Dimmic, 2006). It also plays an important role in modern generative modeling, where one seeks to evaluate whether the model-generated samples resemble the observed data distribution. Consequently, developing powerful and computationally efficient two-sample tests has attracted substantial attention in both theory and practice.

During the past several decades, kernel-based two-sample tests have received considerable attention due to their strong empirical performance in detecting distributional discrepancies in a broad range of settings. The most widely used kernel two-sample test, proposed by Gretton et al. (2006), is based on the MMD. In particular, the MMD between two distributions is defined as the largest difference in expectations over functions belonging to the unit ball of a reproducing kernel Hilbert space (RKHS). Given independent observations $\{X_i^{(1)}\}_{i=1}^{n_1}$ and $\{X_j^{(2)}\}_{j=1}^{n_2}$, an unbiased empirical estimator of $\text{MMD}^2$ is given by

$$
\begin{aligned}
\widehat{\text{MMD}}^2 &= \frac{1}{n_1(n_1-1)} \sum_{i=1}^{n_1} \sum_{j=1, j \neq i}^{n_1} h(X_i^{(1)}, X_j^{(1)}) \\
&+ \frac{1}{n_2(n_2-1)} \sum_{i=1}^{n_2} \sum_{j=1, j \neq i}^{n_2} h(X_i^{(2)}, X_j^{(2)}) \\
&- \frac{2}{n_1 n_2} \sum_{i=1}^{n_1} \sum_{j=1}^{n_2} h(X_i^{(1)}, X_j^{(2)}),
\end{aligned}
$$

where $h(\cdot, \cdot)$ is a positive definite kernel. Based on this statistic, the null hypothesis $H_0$ is rejected whenever $\widehat{\text{MMD}}^2$ exceeds a significance threshold $\tau(\alpha)$, so that the resulting test controls the Type I error at level $\alpha$. In practice, the threshold $\tau(\alpha)$ can be obtained via a permutation-based calibration scheme. Although MMD-based tests perform well in many scenarios, they can exhibit unsatisfactory, and sometimes counterintuitive, behavior under certain common alternatives. A toy example illustrating this phenomenon is presented below.

**Toy Example.** Suppose 100 observations are generated from two distributions, $P_1 = N(0, I_p)$ and $P_2 = N(\mu\mathbf{1}, I_p + \sigma^2\mathbf{1}\mathbf{1}^\top)$, respectively, where $p = 50$, $\mathbf{1} = (1, \cdots, 1)^\top \in \mathbb{R}^p$ and $I_p$ is a $p \times p$ identity matrix. Then we consider three types of alternatives: mean shift

---

[1] School of Statistics and Data Science, Southwestern University of Finance and Economics, Chengdu, China [2] Joint Laboratory of Data Science and Business Intelligence, Southwestern University of Finance and Economics, Chengdu, China [3] Center of Statistical Research, Southwestern University of Finance and Economics, Chengdu, China [4] Laboratory for Data Intelligence and Innovation in Finance and Economics, Southwestern University of Finance and Economics, Chengdu, China. Correspondence to: Yao Yan <yanyao@swufe.edu.cn>.

*Proceedings of the 43rd International Conference on Machine Learning*, Seoul, South Korea. PMLR 306, 2026. Copyright 2026 by the author(s).

($\mu = 0.2, \sigma^2 = 0$), covariance shift ($\mu = 0, \sigma^2 = 0.2$) and location shift ($\mu = 0.2, \sigma^2 = 0.2$). After conducting 500 replications, the estimated power of the MMD test with a permutation-based approach is 0.994, 0.358 and 0.962 for the above three alternatives, respectively. Surprisingly, under the location shift alternative, although there are additional variance differences, the MMD test exhibits slightly lower power than under the mean shift alternative. Besides, the MMD test has low power under the covariance shift alternative.

**Why MMD Fails?** According to the explicit form of $\widehat{\text{MMD}}^2$, this statistic can be interpreted as an aggregate discrepancy between the *within-sample* and *between-sample* kernel similarities across all observations. More precisely, we may write

$$\widehat{\text{MMD}}^2 = \sum_{i=1}^{n} \big(\hat{d}_i^{(in)} - \hat{d}_i^{(bet)}\big),$$

where the explicit expressions of $\hat{d}_i^{(in)}$ and $\hat{d}_i^{(bet)}$ are provided in Section 3.1. This decomposition suggests that the individual contributions $\hat{d}_i^{(in)} - \hat{d}_i^{(bet)}$ may cancel each other under certain alternatives, which could substantially reduce the magnitude of $\widehat{\text{MMD}}^2$ and hence diminish the test power.

To illustrate this phenomenon, we plot the boxplots of $\hat{d}_i^{(in)} - \hat{d}_i^{(bet)}$ for the two samples in a representative replication. As shown in Figure 1, under the mean-shift alternative, the majority of $\hat{d}_i^{(in)} - \hat{d}_i^{(bet)}$ values are positive, yielding a large value of $\widehat{\text{MMD}}^2$ and consequently high power. In contrast, under the covariance-shift alternative, the signs of these differences tend to be opposite across the two samples, which leads to substantial cancellation and results in a smaller $\widehat{\text{MMD}}^2$ and hence low power. A similar cancellation effect also arises under the location-shift alternative. In this case, the differences corresponding to observations from the first sample are typically much larger than zero, whereas those from the second sample become significantly negative. As a result, the contributions from the two samples offset each other, leading to a reduced $\widehat{\text{MMD}}^2$ and lower power compared with the mean-shift setting. Related discussions can also be found in Song & Chen (2024).

**Our Approach.** Motivated by the above analysis, we propose a kernel-based MOD test that is designed to exhibit strong power against a broad class of practically relevant alternatives. The key idea is to maximize the squared discrepancy between the *within-sample* and *between-sample* average kernel similarities across all observations. As illustrated in Figure 2, unlike the classical MMD statistic that aggregates discrepancies globally, MOD takes the maximum over these local departures, making it particularly sensitive to subtle distributional shifts. To further enhance power and

mitigate the dependence on kernel specification, we additionally develop a fused MOD procedure that adaptively aggregates evidence across multiple kernels.

Compared with existing tests introduced in Section 2, the proposed test enjoys the following three appealing advantages.

- **Theoretical guarantee.** We rigorously derive the asymptotic distribution of the MOD test under the null hypothesis, after exploring the complex correlation structures between these differences. This result enables the critical value to be efficiently approximated via numerical methods. Furthermore, we establish the consistency of the MOD test under mild regularity conditions.

- **High power.** To assess the finite-sample behavior of the MOD test, we conduct extensive experiments on both synthetic and real-world datasets. The empirical results align well with our theoretical analysis. In addition, the MOD test demonstrates superior power compared to existing methods under a range of complex shift alternatives.

- **High extendability.** Owing to the simple form of the proposed statistic, the proposed MOD test can be easily extended to address the multi-sample problem. For the extended version, we also derive the asymptotic null distribution and prove its consistency.

**Paper Outline.** The paper is organized as follows. Section 2 reviews related work. Section 3 introduces the MOD test and its theoretical properties. Sections 4–5 develop the fused MOD and its multisample extension. Section 6 presents numerical studies, and Section 7 concludes. Proofs are deferred to the Appendix.

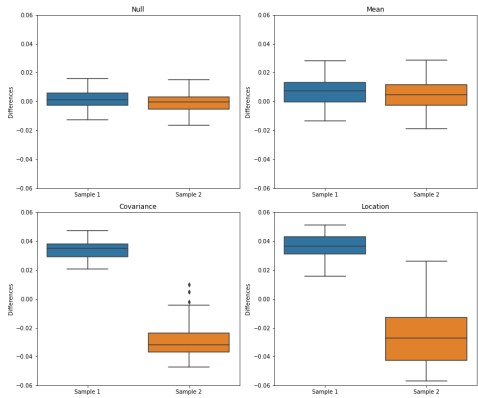

*Figure 1.* Boxplots of $\hat{d}_i^{(in)} - \hat{d}_i^{(bet)}$ ($i = 1, \cdots, n$) for two different samples. Top left: null hypothesis; top right: mean shift alternative; bottom left: covariance shift alternative; bottom right: location shift alternative.

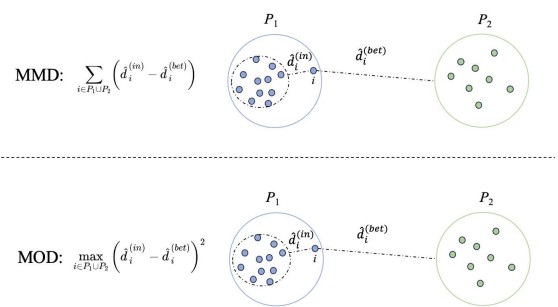

*Figure 2.* Main differences in computing the MMD statistic and the MOD statistic.

## 2. Related Works

In this section, we review the literature on two-sample testing, focusing on MMD-based tests and recent developments in kernel selection and fusion.

**Two-sample Test.** During the past several decades, much effort in the literature has been devoted to addressing the two-sample problem via parametric methods (see, e.g., Lopes et al., 2011; Cai et al., 2014). However, these tests are not applicable when the distribution of observations is misspecified. To avoid this issue, numerous researchers have focused on nonparametric tests, such as the Kolmogorov-Smirnov test and the Wilcoxon test (see, e.g., Bickel, 1969; Oja & Randles, 2004). While these tests are useful and insensitive to distribution assumptions, they are not directly applicable to high-dimensional multivariate data. Recently, several nonparametric tests have been proposed for high-dimensional data, including rank-based tests (see, e.g., Baumgartner et al., 1998; Oja, 2010), interpoint distance-based tests (see, e.g., Székely et al., 2004; Li, 2018), graph-based tests (see, e.g., Friedman & Rafsky, 1979; Chen & Friedman, 2017) and kernel-based tests (see, e.g., Gretton et al., 2006; 2012).

**MMD.** Among kernel-based two-sample tests, the MMD-based tests is one of the most widely used tools for assessing distributional equality (Gretton et al., 2006). Its appeal lies in its ability to detect complex nonlinear differences while enjoying strong theoretical guarantees and being straightforward to implement. To further improve scalability in large-sample settings, several computationally efficient variants have been proposed, including the linear-time MMD (Gretton et al., 2012), the block MMD (Zaremba et al., 2013), and the cross MMD (Shekhar et al., 2022).

Despite these advances, the power of MMD-based tests depends crucially on the choice of kernel. A common strategy is to adopt data splitting, where one subset is used to select or optimize the kernel and the remaining subset is used to perform the final test (Sutherland et al., 2017). Such kernel selection is often carried out in a supervised manner to distinguish the two samples, either by directly learning deep kernels (Sutherland et al., 2017; Liu et al., 2020; 2021) or indirectly through the associated witness function (Kübler et al., 2022). Although effective in practice, this approach may lead to a loss of power since only part of the data is used for testing.

More recently, kernel fusion has been actively investigated through combining multiple kernels without data splitting. A representative line of work is MMD-aggregation, which aggregates a collection of MMD statistics computed under different kernels to construct an adaptive test with theoretical guarantees on power and minimax separation rates (Schrab et al., 2023b). Several extensions have also been proposed to accommodate different computational and statistical considerations, including linear-time estimators (Schrab et al., 2022) and approaches based on spectral regularization (Hagrass et al., 2024). Another prominent method is fused MMD, which performs adaptive kernel fusion via a principled weighting scheme, leading to well-calibrated tests with competitive power and computational efficiency (Biggs et al., 2023). This framework is sufficiently flexible to extend beyond the MMD setting. Motivated by this idea, we further propose the fused MOD test, which adaptively aggregates evidence across multiple kernels within the MOD framework.

**Beyond the MMD.** Beyond MMD, a variety of kernel-based two-sample tests have been developed. For example, Chwialkowski et al. (2015) proposed a linear-time statistic based on the average squared distance between empirical kernel mean embeddings evaluated at $J$ randomly sampled test locations. Building on this framework, Jitkrittum et al. (2016) introduced a related procedure that selects the $J$ locations by maximizing a lower bound on test power. While these methods are well understood in low-dimensional regimes, their behavior in high-dimensional settings remains less explored (Shekhar et al., 2022).

## 3. Kernel-based MOD Test

### 3.1. Notations and Tests

For the sake of convenience, we rearrange the observations sample-by-sample, and denote $\mathcal{Z} = (Z_1, \cdots, Z_n) = (X_1^{(1)}, \cdots, X_{n_1}^{(1)}, X_1^{(2)}, \cdots, X_{n_2}^{(2)})$. In addition, assume that the set $\mathcal{C}_k = \{\sum_{l=0}^{k-1} n_l + 1, \cdots, \sum_{l=0}^{k} n_l\}$ $(k = 1, 2)$ contains the collection of indices associated with the observations coming from the $k$-th sample, where $n_0 = 0$. Moreover, for each $i = 1, \cdots, n$, define $g_i$ as the sample label of observation $Z_i$. To conduct the two-sample problem, we construct the test statistic via the following three steps.

First, for any two observations $Z_i$ and $Z_j$ $(i, j = 1, \cdots, n)$, we calculate the kernel distance $h(Z_i, Z_j)$, where $h(\cdot, \cdot)$ is

a kernel function.

Second, we construct the average distance of the *within-sample* and *between-sample*. Specifically, for any observation $Z_i$, the average distance within the sample is defined as follows:

$$\hat{d}_i^{(in)} = \frac{1}{n_{g_i} - 1} \sum_{j \in \mathcal{C}_{g_i}} h(Z_i, Z_j).$$

Analogously, the average distance between samples is defined as

$$\hat{d}_i^{(bet)} = \frac{1}{n - n_{g_i}} \sum_{j \notin \mathcal{C}_{g_i}} h(Z_i, Z_j).$$

If distributions $P$ and $Q$ are the same, the average distances of the *within-sample* and *between-sample* are quantitatively similar for each observation, and then the difference between $\hat{d}_i^{(in)}$ and $\hat{d}_i^{(bet)}$ approaches 0. This motivates us to propose the test statistic in the third step below.

Third, under the null hypothesis $H_0$, we have $E(\hat{d}_i^{(in)} - \hat{d}_i^{(bet)}) = 0$ and $Var(\hat{d}_i^{(in)} - \hat{d}_i^{(bet)}) = \left( \frac{1}{n_{g_i} - 1} + \frac{1}{n - n_{g_i}} \right)(\Delta_1 - \Delta_2)$, where $\mu = E[h(Z_i, Z_j)]$, $\Delta_1 = E[h(Z_i, Z_j) - \mu]^2$, and $\Delta_2 = E[(h(Z_i, Z_j) - \mu)(h(Z_i, Z_k) - \mu)]$ for any $i \neq j \neq k$. Accordingly, we maximize the squared difference between the two types of average distance across all observations and propose the following test statistic:

$$T = \max_{1 \leq i \leq n} T_i^2, \quad (1)$$

where $T_i = \frac{\hat{d}_i^{(in)} - \hat{d}_i^{(bet)}}{\left\{ \left( \frac{1}{n_{g_i} - 1} + \frac{1}{n - n_{g_i}} \right)(\hat{\Delta}_{i,1} - \hat{\Delta}_{i,2}) \right\}^{1/2}}$, $\hat{\mu}_i = \frac{1}{n-1} \sum_{j \neq i} h(Z_i, Z_j)$, $\hat{\Delta}_{i,1} = \frac{1}{n-1} \sum_{j \neq i} [h(Z_i, Z_j) - \hat{\mu}_i]^2$, and $\hat{\Delta}_{i,2} = \frac{1}{n-1} \frac{1}{n-2} \sum_{j \neq i} \sum_{k \neq j \neq i} [h(Z_i, Z_j) - \hat{\mu}_i][h(Z_i, Z_k) - \hat{\mu}_i]$. Based on the above analysis, the test statistic $T$ is expected to be small under the null hypothesis $H_0$, whereas it is large under the alternative hypothesis $H_1$.

According to (1), the proposed statistic is constructed by maximizing the discrepancy across observations, rather than aggregating discrepancies through summation. Consequently, compared with MMD-based procedures, the proposed MOD test is particularly effective for detecting localized distributional shifts, especially in settings where the discrepancy is driven by only a small subset of observations. This phenomenon is further illustrated in Experiment III of Section C.1.

Additionally, Zhu & Shao (2021) showed that MMD permutation tests are primarily sensitive to sufficiently strong cumulative marginal mean or variance differences, but may exhibit low, or even trivial, power when such marginal differences are weak, particularly when the two distributions share similar marginals and differ mainly in higher-order or dependence structures. Their results provide additional insight into why, in our simulations, MOD tends to outperform MMD under covariance and more general distribution shift alternatives.

Our statistic is also related to the ME test of Chwialkowski et al. (2015), which evaluates distributional discrepancies at a finite number of randomly sampled or carefully selected test locations. In contrast, the proposed statistic evaluates discrepancies across the entire observed sample through a global maximization step. This structure facilitates rigorous theoretical analysis, enabling us to establish validity and effectiveness in both low- and high-dimensional regimes.

*Remark* 3.1. To avoid the cancellation effect involved in MMD, another possible approach is to consider nonnegative sum-type statistics, such as $T_{\text{SAD}} = \sum_{i=1}^n T_i^2$. Compared with this statistics, the proposed MOD statistics is based on max aggregation and is therefore more effective under sparse alternatives, where only a small number of $T_i^2$s take large values. Consequently, it is particularly well suited for detecting subtle distributional differences. More importantly, sum-type statistics are substantially more challenging to analyze theoretically. First, the components $T_i$ exhibit strong dependence, making the limiting distribution of the sum difficult to characterize. Second, after aggregation through summation, the plug-in estimation errors accumulate across coordinates, which complicates the proof that such errors are asymptotically negligible. In contrast, max aggregation leads to a considerably cleaner and more tractable theoretical analysis.

### 3.2. Asymptotic Null Distribution

Before establishing the limiting null distribution of the proposed test statistic $T$, we need to find the covariance matrix of $(T_1, \cdots, T_n)^\top$. To this end, we define

$$Q_i = \alpha(g_i)\big(\hat{d}_i^{(in)} - \hat{d}_i^{(bet)}\big),$$

where $\alpha(g_i) = \left\{ \left( \frac{1}{n_{g_i} - 1} + \frac{1}{n - n_{g_i}} \right)(\Delta_1 - \Delta_2) \right\}^{-1/2}$. Let $\Sigma = (\sigma_{ij}) \in \mathbb{R}^{n \times n}$ be the covariance matrix of $(Q_1, \cdots, Q_n)^\top$. Under $H_0$, we have $\sigma_{ii} = 1$ and $\sigma_{ij}(i \neq j)$ is expressed as

$$\sigma_{ij} = \begin{cases} \alpha^2(g_i) \left[ \frac{\Delta_1 + (n_{g_i} - 4)\Delta_2}{(n_{g_i} - 1)^2} + \frac{\Delta_2}{n - n_{g_i}} \right], & \text{if } g_i = g_j, \\ \alpha(g_i)\alpha(g_j) \frac{\Delta_1 - (n+2)\Delta_2}{n_{g_i} n_{g_j}}, & \text{if } g_i \neq g_j. \end{cases}$$

After carefully investigating the relationship between $Q_i$s and $T_i$s, the asymptotic distribution of $T$ is shown in Theorem 3.2. Before presenting the theorem, we introduce the following condition to facilitate the theoretical analysis.

(C1) There exists some constant $0 < \gamma_1 < 1$ such that $n_1/n \to \gamma_1$ as $n \to \infty$.

Condition (C1) assumes that the number of observations from two samples is approximately balanced. Under this condition, we have the following theorem.

**Theorem 3.2.** *Assume that Condition (C1) holds and that $\Sigma$ is positive definite. Under the null hypothesis $H_0$, for any $x > 0$, we have that*

$$P\{T < x\} - P\{\max_{1 \le i \le n} Z_i^2 < x\} \to 0,$$

*as $n \to \infty$, where $Z = (Z_1, \cdots, Z_n)^\top$ is a multivariate normally distributed random variable with mean zero and covariance matrix $\Sigma$.*

Theorem 3.2 remains valid in both fixed-$p$ and diverging-$p$ settings. When $p \to \infty$, additional dimension-dependent constants may arise in the plug-in error bounds and concentration bounds in the proof of Step 3 of Theorem 3.2. However, since the analysis is based on bounded kernel functions, the relevant concentration bounds can be controlled uniformly with respect to the dimension. Consequently, the same argument applies to both low- and high-dimensional regimes. To the best of our knowledge, analogous results are generally unavailable for the classical MMD framework (Zhu & Shao, 2021).

To ensure the positive definiteness of the matrix $\Sigma$ and the validity of the associated normalization term, the kernel function, particularly its bandwidth parameter, must be chosen such that $\Delta_1 - \Delta_2$ remains bounded away from zero. If the bandwidth is chosen excessively large or small, the kernel function may become nearly constant or degenerate, thereby invalidating the proposed test. Therefore, in high-dimensional settings, the bandwidth should be appropriately scaled with the data dimension. In practice, we recommend a data-driven bandwidth selection procedure based on quantiles of the pairwise distances.

Based on Theorem 3.2, we can reject the null hypothesis at the significant level $\alpha$ if $T \ge q_\alpha$, where $q_\alpha$ is the $(1 - \alpha)$-th quantile of the distribution $\max_{1 \le i \le n} Z_i^2$. Since this distribution does not have a closed form, we can employ a numerical method to find the critical value. Specifically, we can generate $M$ observations from the multivariate normal distribution with mean zero and covariance matrix $\hat{\Sigma}$, where $\hat{\Sigma}$ is the estimator of $\Sigma$ by replacing $\Delta_1$ and $\Delta_2$ with $\hat{\Delta}_1 = \frac{1}{n(n-1)} \sum_{i=1}^n \sum_{j \neq i} [h(Z_i, Z_j) - \hat{\mu}]^2$, and $\hat{\Delta}_2 = \frac{1}{n(n-1)(n-2)} \sum_{i=1}^n \sum_{j \neq i} \sum_{k \neq j \neq i} [h(Z_i, Z_j) - \hat{\mu}] \cdot [h(Z_i, Z_k) - \hat{\mu}]$, where $\hat{\mu} = \frac{1}{n(n-1)} \sum_{i=1}^n \sum_{j \neq i} h(Z_i, Z_j)$. Denote the generated $m$-th observation by $B^{(m)} = (B_1^{(m)}, \cdots, B_n^{(m)})^\top$ for $m = 1, \cdots, M$. Subsequently, we compute $S^{(m)} = \max_{1 \le i \le n} B_i^{(m)2}$, and obtain the empirical critical value $S_\alpha$, which is the $(1 - \alpha)$-th quantile of

$S^{(1)}, \cdots, S^{(M)}$. Finally, we compare the test statistic $T$ with the empirical critical value $S_\alpha$, and draw the appropriate conclusion. For simplicity, we refer to this test as the bootstrap MOD test (MODboot) hereafter. Note that the above procedure relies on estimating $\Sigma$ and the estimation step may introduce additional variability. However, it can be shown that the estimation error has only a limited impact on the finite-sample power of the test.

Although the proposed MOD test involves maximization over all observations together with covariance matrix estimation, its computational complexity remains tractable. Specifically, for the statistic in (1), the quantities $\hat{\mu}_i, \hat{\Delta}_{i,1}$, and $\hat{\Delta}_{i,2}$ can each be computed in $O(n)$ time via row-wise summation for every $i$. Consequently, evaluating the full test statistic over $i = 1, \ldots, n$ requires an overall computational cost of $O(n^2)$. The covariance estimation step has the same order of complexity. Furthermore, the empirical results reported in Appendix C.1 demonstrate that the computational cost of the proposed method remains practically manageable even for relatively large sample sizes.

### 3.3. Consistency

In this subsection, we show the consistency of the tests discussed above. To this end, we investigate the behavior of $\hat{\Delta}_{i,1}$ and $\hat{\Delta}_{i,2}$ involved in (1) under the alternative hypothesis $H_1$. For any $i \in \mathcal{C}_k$, $j \in \mathcal{C}_l$ and $m \in \mathcal{C}_s$, define $\mu_{kl} = E(h(X_i, X_j))$, $\Delta_{1kl} = E[(h(X_i, X_j) - \mu_{kl})^2]$, and $\Delta_{2kls} = E[(h(X_i, X_j) - \mu_{kl})(h(X_i, X_m) - \mu_{ks})]$. We then show the asymptotic properties of the means of $\hat{\Delta}_{i,1}$ and $\hat{\Delta}_{i,2}$ below.

**Proposition 3.3.** *Under $H_1$, we have $E(\hat{\mu}_i) \to \gamma_1 \mu_{g_i 1} + \gamma_2 \mu_{g_i 2}$, $E(\hat{\Delta}_{i,1}) \to \Delta_{i,1}$ and $E(\hat{\Delta}_{i,2}) \to \Delta_{i,2}$, where the explicit expressions of $\Delta_{i,1}$ and $\Delta_{i,2}$ are stated in the Appendix A.2.*

Accordingly, the differences between the *within-sample* and *between-sample* distances for each observation $i$ can be measured by

$$\nu_i = \frac{(\mu_{g_i}^{(in)} - \mu_{g_i}^{(bet)})^2}{\left(\frac{1}{n_{g_i} - 1} + \frac{1}{n - n_{g_i}}\right)(\Delta_{i,1} - \Delta_{i,2})}, \qquad (2)$$

where $\mu_{g_i}^{(in)} = \mu_{g_i g_i}$ and $\mu_{g_i}^{(bet)} = \mu_{g_i k}$ $(k \neq g_i)$. Note that $\mu_{g_i}^{(in)} - \mu_{g_i}^{(bet)}$ quantifies the discrepancy between the within-sample and between-sample distances associated with observation $i$. Accordingly, $\nu_i = 0$ for all $i$ under $H_0$, whereas $\nu_i > 0$ for some $i$ under $H_1$. Moreover, $\nu_i$ tends to increase as the separation between the two distributions becomes more pronounced. Therefore, $\nu_i$ can be viewed as a measure of signal strength. Using Proposition 3.3 and (2), we are able to show the consistency of the MOD test below.

**Theorem 3.4.** *Assume that Condition (C1) holds and that $\Sigma$ is positive definite. Under the alternative hypothesis $H_1$, if $\max_{1 \leq i \leq n} \nu_i > (4 + \epsilon) \log n$ for some positive constant $\epsilon$, then we have*

$$P(T > S_\alpha) \to 1,$$

*where $S_\alpha$ is the empirical critical value.*

Theorem 3.4 implies that the power of the proposed test converges to 1. To guarantee the consistency of the proposed test, Theorem 3.4 assumes that the maximum value of $\nu_i$ is not too small. This assumption is mild because it imposes a requirement on the strength of the difference between two samples. More importantly, in contrast to many common tests that require conditions on the average signal strength across observations, the proposed MOD test only relies on the maximum value of $\nu_i$. This feature makes the proposed procedure particularly effective for detecting subtle or localized distributional differences.

## 4. Kernel-based fused MOD Test

To compute the MOD test statistic $T$, we employ a kernel function to quantify discrepancies between observations. In practice, however, the optimal choice of the kernel family (e.g., Gaussian or Laplace) and the associated bandwidth parameter is typically unknown. A large body of literature has shown that the finite-sample performance of kernel-based two-sample tests can be highly sensitive to such choices. This motivates us to incorporate data-adaptive kernel selection into the MOD framework in order to improve robustness and enhance power.

To this end, we build upon the kernel fusion idea of Biggs et al. (2023) and propose a fused version of MOD, which adaptively aggregates evidence across multiple kernels without data splitting. Given a candidate kernel set $\mathcal{H} = \{h_1, \ldots, h_H\}$, we compute the kernel-specific statistics $\{T(h_i)\}_{i=1}^H$, where $T(h_i)$ denotes the MOD statistic evaluated under kernel $h_i$. We then aggregate these statistics via a soft-maximum operator,

$$T_{\text{agg}} = \frac{1}{\lambda} \log \left( \frac{1}{H} \sum_{i=1}^H \exp\big(\lambda T(h_i)\big) \right),$$

where $\lambda > 0$ controls the smoothness of the aggregation. As $\lambda$ increases, $T_{\text{agg}}$ approaches $\max_{1 \leq i \leq H} T(h_i)$, while smaller values yield a smoother fusion that pools evidence across multiple kernels. This aggregation inherits the main strengths of kernel fusion methods, while, in contrast to MMD-based fusion procedures that often require additional studentisation to align scales, our MOD statistics are already properly normalised by construction (see (1)), and thus are not affected by scale mismatch across kernels or bandwidth choices.

To determine the significance threshold, we adopt a permutation-based procedure. By Lemma A.1, valid type-I error control is preserved even when the kernel set $\mathcal{H}$ is selected in a data-dependent manner, as long as the selection rule is permutation-invariant and depends only on the unordered pooled sample. Accordingly, for each permutation $b = 1, \ldots, B$, we randomly permute the sample labels and recompute the aggregated statistic $T_{\text{agg}}^{(b)}$. The critical value is then obtained as the empirical $(1 - \alpha)$ quantile of $\{T_{\text{agg}}^{(b)}\}_{b=1}^B$, and we reject the null hypothesis whenever the observed statistic $T_{\text{agg}}$ exceeds this threshold.

## 5. Extension to K-sample problem

In this subsection, we extend the proposed test to address the K-sample problem. Before stating the proposed test, we first introduce some notations. Let $X_i^{(k)} \in \mathbb{R}^p$ $(i = 1, \cdots, n_k)$ be the independent and identically distributed observations from $P_k$ $(k = 1, \cdots, K)$, and the total number of observations is $n = \sum_{k=1}^K n_k$. The definitions of $\mathcal{Z}$, $Z_i$, $\mathcal{C}_k$ and $g_i$ are the same as those in Section 3.1. To compare the $K$-sample distributions, we consider the following hypotheses:

$$H_0 : P_1 = \cdots = P_K \quad \text{versus}$$

$$H_1 : P_{k_1} \neq P_{k_2} \text{ for some } 1 \leq k_1 < k_2 \leq K.$$

Analogue to Section 3.1, for every observation $i$, we define the average distance within the sample as $\hat{d}_{\tau,i}^{(in)} = \frac{1}{n_{g_i}-1} \sum_{j \in \mathcal{C}_{g_i}} h(Z_i, Z_j)$, and define the average distance between samples as $\hat{d}_{\tau,i}^{(bet)} = \frac{1}{n-n_{g_i}} \sum_{j \notin \mathcal{C}_{g_i}} h(Z_i, Z_j)$. Then we propose the following statistic:

$$T_\tau = \max_{1 \leq i \leq n} T_{\tau,i}^2, \tag{3}$$

where $T_{\tau,i} = \frac{\hat{d}_{\tau,i}^{in} - \hat{d}_{\tau,i}^{bet}}{\left\{ \left( \frac{1}{n_{g_i}-1} + \frac{1}{n-n_{g_i}} \right) \left[ \hat{\Delta}_{\tau,i}^{(1)} - \hat{\Delta}_{\tau,i}^{(2)} \right] \right\}^{1/2}}$, $\hat{\mu}_{\tau,i} = \frac{1}{n-1} \sum_{i \neq j} h(Z_i, Z_j)$, $\hat{\Delta}_{\tau,i}^{(1)} = \frac{1}{n-1} \sum_{i \neq j} [h(Z_i, Z_j) - \hat{\mu}_{\tau,i}]^2$, and $\hat{\Delta}_{\tau,i}^{(2)} = \frac{1}{n-1} \frac{1}{n-2} \sum_{i \neq j} \sum_{j \neq m \neq i} [h(Z_i, Z_j) - \hat{\mu}_{\tau,i}][h(Z_i, Z_m) - \hat{\mu}_{\tau,i}]$. According to the expression of (3), we expect that $T_\tau$ is small under the null hypothesis, whereas it is large under the alternative hypotheses. Using similar techniques as those in Sections 3.2 and 3.3, we derive the asymptotic distribution of $T_\tau$ and establish the consistency of the proposed tests as follows.

(C2) For any $1 \leq k \leq K$, there exists some constant $\gamma_k$ such that $n_k/n \to \gamma_k$ as $n \to \infty$. In addition, there exist two constants $\gamma_{\min}$ and $\gamma_{\max}$ such that $0 < \gamma_{\min} \leq \gamma_k \leq \gamma_{\max} < 1$.

**Theorem 5.1.** *Assume that Condition (C2) holds and that $\Sigma_\tau$ is positive definite, with $\Sigma_\tau$ defined in the Appendix B.*

*Under the null hypothesis, for any $x > 0$, we have that*

$$P\{T_\tau < x\} - P\{\max_{1 \le i \le n} Z_i^2 < x\} \to 0,$$

*as $n \to \infty$, where $Z = (Z_1, \cdots, Z_n)^\top$ is a multivariate normally distributed random variable with mean zero and covariance matrix $\Sigma_\tau$.*

**Theorem 5.2.** *Assume that Condition (C2) holds and that $\Sigma_\tau$ is positive definite. Under the alternative hypothesis, if $\max_{1 \le i \le n} \nu_{\tau,i} > (4 + \epsilon) \log n$ for some positive constant $\epsilon$, then we have*

$$P(T_\tau > S_\alpha) \to 1,$$

*where $S_\alpha$ is the empirical critical value and the specific expression of $\nu_{\tau,i}$ is defined in the Appendix B.*

## 6. Experiments

To evaluate the finite-sample behavior of the proposed procedure, we perform various experiments on both synthetic and real-world datasets. In the following experiments, `MODboot` denotes the MOD test implemented with the median heuristic, whereas `MOD-fuse` refers to its fused variant. Here, the median heuristic refers to selecting the bandwidth of the kernel function as the median of all pairwise distances computed from the pooled observations. For comparison, we additionally report the results of several representative two-sample tests, including the classical MMD with the median heuristic (Gretton et al., 2012, `MMD-median`), the optimized MMD based on data splitting (Sutherland et al., 2017, `MMD-split`), the fused MMD (Biggs et al., 2023, `MMD-fuse`), the aggregated MMD (Schrab et al., 2023a, `MMD-agg`), the efficient aggregated MMD (Schrab et al., 2022, `MMD-aggInc`), and the analytic mean embedding test (Jitkrittum et al., 2016, `ME`).

To construct the collection of candidate kernels, we follow Schrab et al. (2023b) and consider a finite family of Gaussian kernels parameterized by bandwidths $\gamma > 0$,

$$h_\gamma(x, y) = \exp\left(-\frac{\|x - y\|_2^2}{2\gamma^2}\right).$$

Specifically, ten bandwidth candidates are selected via a data-driven grid: we uniformly discretize the interval ranging from one half of the $5\%$ quantile to twice the $95\%$ quantile (for robustness) of the pairwise distances. For `MOD-fuse` and `MMD-fuse`, we set $\lambda = \sqrt{n_{\min}(n_{\min} - 1)}$, where $n_{\min} = \min(n_1, n_2)$. We also examine the robustness of the proposed method to the choice of kernel and move the corresponding results to Appendix C.2 due to space limitations. Throughout all experiments, we set the nominal significance level to 0.05. The code for reproducing all experiments is publicly available at https://github.com/Danpu666/Kernel-based-MOD-Test.

### 6.1. Synthetic data

**Size control.** In Experiment I, we examine the empirical size of all competing tests under the null hypothesis across different sample sizes. Specifically, we generate two samples of equal size $n_1 = n_2 = n/2$ from a $p$-dimensional $t$ distribution with 10 degrees of freedom, where the dimension is fixed at $p = 10$ and the total sample size $n$ ranges from 100 to 300. Then, each test is applied to the simulated data. We repeat this procedure 200 times and estimate the empirical size by the average rejection rate. The results are reported in Figure 3. As shown in Figure 3, the `ME` test exhibits noticeable size distortions especially when the sample size is small. In contrast, the proposed MOD test, along with the remaining methods, maintains rejection rates close to the nominal level for all values of $n$. These findings provide numerical evidence for Theorem 3.2.

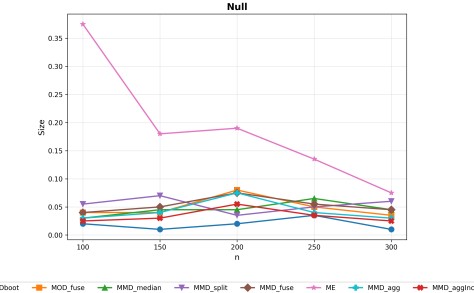

*Figure 3.* Test size vs. number of observations in Experiment I.

**Test power vs. number of observations.** In Experiment II, we compare the empirical power of different tests as the sample size varies. To provide a comprehensive and fair evaluation, we consider four alternatives, including mean shift, covariance shift, location shift, and distribution shift. Specifically, we generate two independent samples from distributions $P$ and $Q$, respectively, where the specifications of $P$ and $Q$ are summarized in Table 1. All other experimental settings are kept the same as in Experiment I.

Figure 4 reports the empirical power of all competing methods across different values of $n$. Except for the `ME` test, both the proposed MOD-based tests and the other procedures exhibit steadily increasing power as $n$ grows under all alternatives, which is consistent with the asymptotic result in Theorem 3.2. Under the mean-shift alternative, our method is slightly less powerful than several competing approaches. However, under covariance-shift and distribution-shift alternatives, the proposed `MOD-fuse` test achieves substantially higher power than the remaining methods. These results suggest that the proposed MOD-based tests are particularly effective for detecting subtle distributional differences.

**Test power vs. strength of signals.** In Experiment III, we further examine the robustness of the proposed methods as the signal strength varies. Specifically, we generate two

samples of equal size $n_1 = n_2 = 75$ from distributions $P$ and $Q$, respectively, with specifications given in Table 1. We fix the dimension at $p = 200$ and vary the signal strength parameter $\delta$, while keeping all other settings identical to those in Experiment I.

The results are presented in Figure 5. Across all alternatives, the power of most methods increases as $\delta$ becomes larger, except for the `ME` test. This pattern is expected, since a larger $\delta$ amplifies the discrepancy between $P$ and $Q$ and thus makes the two distributions easier to distinguish. Under mean-shift alternatives, the proposed `MODboot` and `MOD-fuse` tests are slightly less powerful than some competing approaches. However, under covariance-shift and location-shift alternatives, both `MODboot` and `MOD-fuse` consistently outperform the remaining methods, echoing the findings in Experiment II. These findings suggest that the proposed MOD-based tests are particularly well suited for detecting localized departures from the null, especially when the distributional discrepancy is driven by only a small fraction of observations.

### 6.2. Real Data

We investigate the empirical performance of different two-sample tests using the MNIST handwritten digit dataset. MNIST is a widely used benchmark in image analysis, consisting of grayscale images of handwritten digits with a resolution of $28 \times 28$ pixels. Following the experimental protocol of Schrab et al. (2023b), we design five classification experiments with increasing levels of difficulty. Specifically, we fix one reference distribution $P$ and consider five alternative distributions $Q_i$ $(i = 1, \ldots, 5)$, whose digit compositions are summarized in Table 2. For each experiment, we randomly draw $n/2$ observations from $P$ and $n/2$ observations from $Q_i$, and apply eight competing two-sample tests to distinguish between the two distributions. The total sample size $n$ ranges from 40 to 2000, and the entire procedure is repeated 200 times to estimate the empirical power of each method.

Figure 6 reports the empirical power of all competing tests. As expected, the power of all methods increases monotonically with the sample size $n$. When distinguishing $P$ from $Q_i$ for $i = 1, \ldots, 4$, both the proposed test and the competing methods exhibit strong performance when the sample size is large. In contrast, the power of all methods is substantially lower in the fifth experiment than in the first four experiments. This observation is expected, as the fifth experiment represents the most challenging setting, where the distributions $P$ and $Q_5$ differ only marginally. Notably, the proposed `MOD-fuse` test consistently outperforms the competing approaches in this difficult scenario and achieves power close to one when the sample size is sufficiently large. This result highlights the advantage of the proposed

method in detecting subtle distributional differences in high-dimensional image data.

*Table 2.* The construction of sets $P$ and $Q_i$.

| Set | Digits |
| --- | --- |
| $P$ | 0,1,2,3,4,5,6,7,8,9 |
| $Q_1$ | 1,3,5,7,9 |
| $Q_2$ | 0,1,3,5,7,9 |
| $Q_3$ | 0,1,2,3,5,7,9 |
| $Q_4$ | 0,1,2,3,4,5,7,9 |
| $Q_5$ | 0,1,2,3,4,5,6,7,9 |

## 7. Conclusion

This paper proposes a kernel-based MOD test for the two-sample problem. The proposed statistic maximizes, over all observations, the squared discrepancy between the average within-sample and between-sample kernel similarities. We rigorously derive the asymptotic null distribution of the MOD test and establish its consistency under general alternatives. To enhance robustness and power in practice, we further develop a fused MOD procedure. Extensive experiments on both synthetic and real-world datasets demonstrate that the proposed MOD-based tests consistently outperform existing two-sample testing methods.

## Acknowledgements

We thank the anonymous reviewers for helpful comments on earlier versions of this paper. No potential conflict of interest was reported by the authors. Dan Pu's research is supported by the National Natural Science Foundation of China (72403202). Wei Lan's research was supported by the National Key R&D Program of China (2022YFA1003702), the National Natural Science Foundation of China (72422020, 72333001, 12531011) , the Joint Lab of Data Science and Business Intelligence at Southwestern University of Finance and Economics.

## Impact Statement

This paper presents work whose goal is to advance the field of Machine Learning. There are many potential societal consequences of our work, none which we feel must be specifically highlighted here.

*Table 1.* Specifications of $P$ and $Q$. Set vector $\Delta = (\underbrace{1, \cdots, 1}_{p/2}, \underbrace{0, \cdots, 0}_{p/2}) \in \mathbb{R}^p$ and matrix $\Theta = (\theta_{ij}) \in \mathbb{R}^{p \times p}$ with $\theta_{ij} = 0.4^{|i-j|}$.

| Experiment | Type | $P$ | $Q$ |
|---|---|---|---|
| I | Null | $t_{10}(\mathbf{0}_p, \Theta)$ | $t_{10}(\mathbf{0}_p, \Theta)$ |
| II | Mean | $t_{10}(\mathbf{0}_p, \Theta)$ | $t_{10}(0.35\Delta, \Theta)$ |
| | Covariance | $t_{10}(\mathbf{0}_p, \Theta)$ | $t_{10}(\mathbf{0}_p, 1.35\Theta)$ |
| | Location | $t_{10}(\mathbf{0}_p, \Theta)$ | $t_{10}(0.125\Delta, 1.25\Theta)$ |
| | Distribution | $t_{10}(\mathbf{0}_p, \Theta)$ | $t_3(\mathbf{0}_p, \Theta)$ |
| III | Mean | $N(\mathbf{0}_p, I_p)$ | $0.9N(\mathbf{0}_p, I_p) + 0.1N(\delta\mathbf{1}_p, I_p)$ |
| | Covariance | $N(\mathbf{0}_p, I_p)$ | $0.9N(\mathbf{0}_p, I_p) + 0.1N(\mathbf{0}_p, (1+\delta)I_p)$ |
| | Location | $N(\mathbf{0}_p, I_p)$ | $0.9N(\mathbf{0}_p, I_p) + 0.1N(\frac{\delta}{2}\mathbf{1}_p, (1+\delta)I_p)$ |

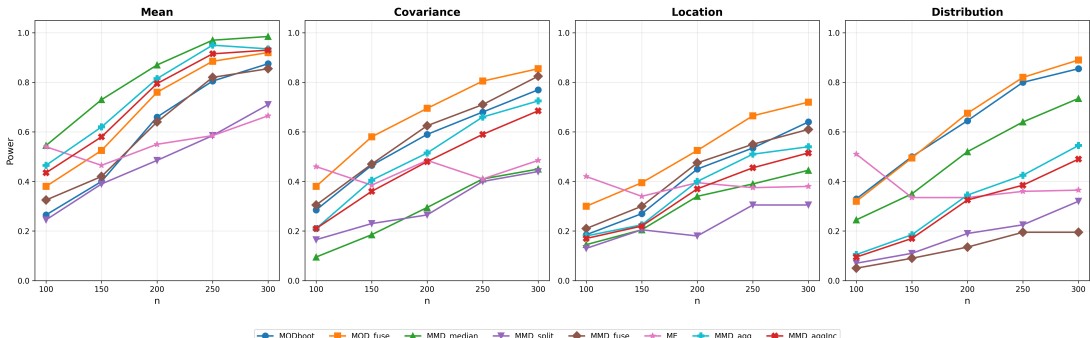

*Figure 4.* Test power vs. number of observations in Experiment II.

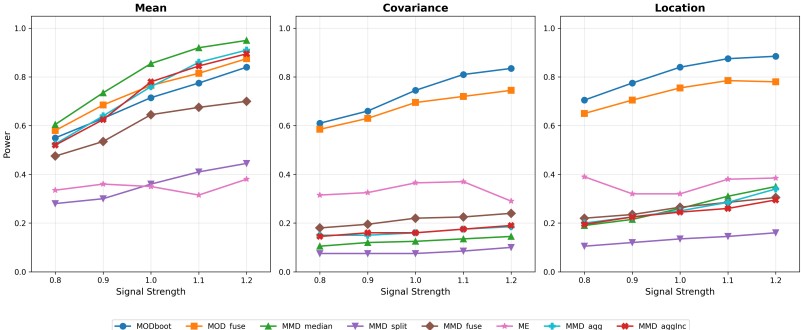

*Figure 5.* Test power vs. signal in Experiment III.

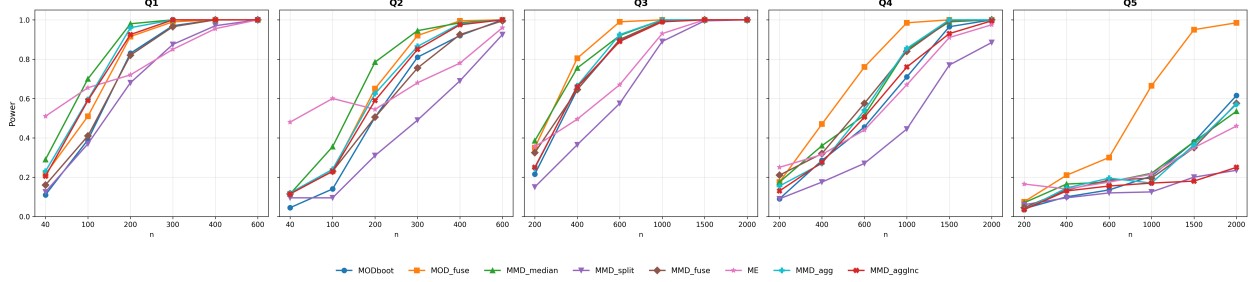

*Figure 6.* Test power vs. number of observations in MNIST dataset.

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

# Appendix

This appendix includes three sections. Section A provides some technique proofs for Proposition 3.3, Theorem 3.2 and Theorem 3.4. Section B extends the MOD test to address multi-sample problem. Section C presents additional experimental results.

## A. Technique Proofs

### A.1. Some useful Lemmas

**Lemma A.1.** *Let $\mathcal{Z} = (Z_1, \ldots, Z_N)$ denote the pooled sample, and let $\mathcal{G}$ denote the permutation group on $\{1, \ldots, N\}$. For each permutation $\bar{g} \in \mathcal{G}$, define*

$$\bar{g}\mathcal{Z} = (Z_{\bar{g}(1)}, \ldots, Z_{\bar{g}(N)}),$$

*namely, the sample obtained by permuting the observations according to $\bar{g}$. Suppose that under the null hypothesis $H_0$, $\bar{g}\mathcal{Z} \overset{d}{=} \mathcal{Z}, \forall \bar{g} \in \mathcal{G}$.*

*Let $\phi(\mathcal{Z})$ be an arbitrary real-valued test statistic. Further, let $G_1, \ldots, G_B$ be independently sampled permutations from $\mathcal{G}$ under the uniform distribution, and define $G_{B+1}\mathcal{Z} = \mathcal{Z}$, so that the original sample is always included among the permuted samples. For any $q \in (0,1)$, define the empirical quantile*

$$Q_{1-q} = \inf\left\{t \in \mathbb{R} : \frac{1}{B+1}\sum_{b=1}^{B+1}\mathbf{1}\{\phi(G_b\mathcal{Z}) \leq t\} \geq 1-q\right\}.$$

*Then the permutation test $\varphi(\mathcal{Z}) = \mathbf{1}\{\phi(\mathcal{Z}) > Q_{1-\alpha}\}$ satisfies*

$$\mathbb{P}_{H_0}\big(\varphi(\mathcal{Z}) = 1\big) \leq \alpha.$$

*That is, the permutation test controls the type-I error at level $\alpha$.*

*Proof.* This lemma follows directly from Theorem 1 of Biggs et al. (2023). $\qquad\square$

Let

$$F_\beta(z) = \beta^{-1}\log\left(\sum_{j=1}^{p}e^{\beta z_j}\right), \qquad z = (z_1, \ldots, z_p)^\top \in \mathbb{R}^p,$$

denote the smooth maximum function with smoothing parameter $\beta > 0$.

Define $m(z) = g(F_\beta(z))$, where $g \in C_b^3(\mathbb{R})$ and $C_b^3(\mathbb{R})$ denotes the class of three-times continuously differentiable functions whose derivatives up to order three are uniformly bounded. For $k = 1, 2, 3$, define $G_k = \sup_{t \in \mathbb{R}}|g^{(k)}(t)|$.

Further, define

$$\pi_j(z) = \frac{e^{\beta z_j}}{\sum_{m=1}^{p}e^{\beta z_m}}, \qquad j = 1, \ldots, p,$$

and let $\delta_{jk}$ denote the Kronecker delta.

**Lemma A.2.** *For every $1 \leq j, k, l \leq p$,*

$$|\partial_j\partial_k m(z)| \leq U_{jk}(z), \ |\partial_j\partial_k\partial_l m(z)| \leq U_{jkl}(z),$$

*where*

$$W_{jk}(z) = \pi_j(z)\delta_{jk} - \pi_j(z)\pi_k(z),$$

$$Q_{jkl}(z) = \pi_j(z)\delta_{jl}\delta_{jk} - \pi_j(z)\pi_l(z)\delta_{jk} - \pi_j(z)\pi_k(z)(\delta_{jl} + \delta_{kl}) + 2\pi_j(z)\pi_k(z)\pi_l(z),$$

$$U_{jk}(z) = G_2\pi_j(z)\pi_k(z) + G_1\beta W_{jk}(z),$$

$$U_{jkl}(z) = G_3\pi_j(z)\pi_k(z)\pi_l(z) + G_2\beta\big(W_{jk}(z)\pi_l(z) + W_{jl}(z)\pi_k(z) + W_{kl}(z)\pi_j(z)\big) + G_1\beta^2 Q_{jkl}(z).$$

*Moreover,*

$$\sum_{j,k=1}^{p} U_{jk}(z) \leq G_2 + 2G_1\beta, \quad \sum_{j,k,l=1}^{p} U_{jkl}(z) \leq G_3 + 6G_2\beta + 6G_1\beta^2.$$

*Proof.* This lemma follows directly from Lemma A.5 of Chernozhukov et al. (2013). □

**Lemma A.3.** *For every $z, w \in \mathbb{R}^p$ satisfying $\max_{1 \leq j \leq p} |w_j|\beta \leq 1$, and $t \in [0,1]$, we have*

$$U_{jk}(z) \lesssim U_{jk}(z+tw) \lesssim U_{jk}(z), \text{ and } U_{jkl}(z) \lesssim U_{jkl}(z+tw) \lesssim U_{jkl}(z),$$

*uniformly over all $1 \leq j, k, l \leq p$.*

*Proof.* This lemma follows directly from Lemma A.6 of Chernozhukov et al. (2013). □

**Lemma A.4.** *Let $V = (V_1, \ldots, V_p)^\top$ and $Y = (Y_1, \ldots, Y_p)^\top$ be centered Gaussian random vectors in $\mathbb{R}^p$ with covariance matrices $\Sigma^V = (\Sigma_{jk}^V)_{1 \leq j,k \leq p}$ and $\Sigma^Y = (\Sigma_{jk}^Y)_{1 \leq j,k \leq p}$, respectively. Define $\Delta_0 = \max_{1 \leq j,k \leq p} \left|\Sigma_{jk}^V - \Sigma_{jk}^Y\right|$. Suppose that there exist constants $0 < c_1 < C_1 < \infty$ such that $\bar{\sigma} := \max_{1 \leq j \leq p} E[Y_j^2] \leq C_1$, and $b_p := E[\max_{1 \leq j \leq p} Y_j] \geq c_1\sqrt{\log p}$. Then there exist constants $c > 0$ and $C > 0$, depending only on $c_1$ and $C_1$, such that*

$$\sup_{t \in \mathbb{R}} \left| P\left(\max_{1 \leq j \leq p} V_j \leq t\right) - P\left(\max_{1 \leq j \leq p} Y_j \leq t\right) \right|$$

$$\leq C\Delta_0^{1/3}\left\{1 \vee \log\left(\frac{p}{\Delta_0}\right)\right\}^{2/3} + Cp^{-c}\sqrt{1 \vee \log\left(\frac{p}{\Delta_0}\right)}.$$

*Proof.* This lemma follows directly from Lemma J.2 of Chernozhukov et al. (2013). □

For a real-valued random variable $\xi$, define its Lévy concentration function by

$$\mathcal{L}(\xi, \varepsilon) = \sup_{x \in \mathbb{R}} P(|\xi - x| \leq \varepsilon), \qquad \varepsilon > 0.$$

**Lemma A.5.** *Let $X = (X_1, \ldots, X_p)^\top$ be a centered Gaussian random vector in $\mathbb{R}^p$ with $\sigma_j^2 = E[X_j^2] > 0$, for $j = 1, \ldots, p$. Define $\underline{\sigma} = \min_{1 \leq j \leq p} \sigma_j$, $\bar{\sigma} = \max_{1 \leq j \leq p} \sigma_j$, and $a_p = E\left[\max_{1 \leq j \leq p} \frac{X_j}{\sigma_j}\right]$.*

*Then the following statements hold.*

*(i) If all variances are equal, namely $\underline{\sigma} = \bar{\sigma} = \sigma$, then for every $\varepsilon > 0$,*

$$\mathcal{L}\left(\max_{1 \leq j \leq p} X_j, \varepsilon\right) \leq \frac{4\varepsilon(a_p + 1)}{\sigma}.$$

*(ii) If the variances are not necessarily equal, namely $\underline{\sigma} < \bar{\sigma}$, then for every $\varepsilon > 0$,*

$$\mathcal{L}\left(\max_{1 \leq j \leq p} X_j, \varepsilon\right) \leq C\varepsilon\left\{a_p + \sqrt{1 \vee \log\left(\frac{\bar{\sigma}}{\varepsilon}\right)}\right\},$$

*where $C > 0$ is a constant depending only on $\underline{\sigma}$ and $\bar{\sigma}$.*

*Proof.* This lemma follows directly from Theorem 3 of Chernozhukov et al. (2015). □

### A.2. The proof of Proposition 3.3

*Proof.* After straightforward calculations, we have that $E(\hat{d}_i^{(in)}) = \mu_{g_i g_i}$, $E(\hat{d}_i^{(out)}) = \mu_{g_i k}$ $(k \neq g_i)$,

$$E(\hat{\mu}_i) = \frac{(n_{g_i} - 1)\mu_{g_i g_i} + (n - n_{g_i})\mu_{g_i k}}{n - 1} \to \gamma_1\mu_{g_i 1} + \gamma_2\mu_{g_i 2}.$$

Employing the above results, we further obtain that

$$
E(\hat{\Delta}_{i,1}) = \frac{1}{n-1} \sum_j E[h(X_i, X_j) - \hat{\mu}_i]^2
$$

$$
= \frac{1}{n-1} \sum_j E\Big\{ [h(X_i, X_j) - \mu_{g_i g_j}]^2 + \mu_{g_i g_j}^2
$$

$$
+ \hat{\mu}_i^2 - 2\hat{\mu}_i \mu_{g_i g_j} + 2\mu_{g_i g_j}[h(X_i, X_j) - \mu_{g_i g_j}] - 2h(X_i, X_j)\hat{\mu}_i + 2\mu_{g_i g_j}\hat{\mu}_i \Big\}
$$

$$
= \frac{1}{n-1} \sum_j \Big\{ \Delta_{1 g_i g_j} + \mu_{g_i g_j}^2 + \frac{1}{(n-1)^2} \sum_{r_1, r_2 \neq i} (\Delta_{2 g_i g_{r_1} g_{r_2}} + \mu_{g_i g_{r_1}} \mu_{g_i g_{r_2}})
$$

$$
- \frac{2}{n-1} \sum_{r \neq i} (\Delta_{2 g_i g_j g_r} + \mu_{g_i g_j} \mu_{g_i g_r}) \Big\}
$$

$$
\to \Delta_{i,1} := \sum_{k \in \{1,2\}} \gamma_k (\Delta_{1 g_i k} + \mu_{g_i k}^2) - \sum_{k,l \in \{1,2\}} \gamma_k \gamma_l (\Delta_{2 g_i kl} + \mu_{g_i k} \mu_{g_i l}), \text{ and}
$$

$$
E(\hat{\Delta}_{i,2}) = \frac{1}{n-1} \frac{1}{n-2} \sum_{j \neq i} \sum_{m \neq j \neq i} E\Big\{ [h(X_i, X_j) - \hat{\mu}_i][h(X_i, X_m) - \hat{\mu}_i] \Big\}
$$

$$
= \frac{1}{n-1} \frac{1}{n-2} \sum_{j \neq i} \sum_{m \neq j \neq i} E\Big\{ [h(X_i, X_j) - \mu_{kl}][h(X_i, X_m) - \mu_{ks}]
$$

$$
+ \mu_{kl} \mu_{ks} - [h(X_i, X_j) + h(X_i, X_m)]\hat{\mu}_i + \hat{\mu}_i^2 \Big\}
$$

$$
= \frac{1}{n-1} \frac{1}{n-2} \sum_{j \neq i} \sum_{m \neq j \neq i} \Big\{ \Delta_{2 g_i g_j g_m} + \mu_{g_i g_j} \mu_{g_i g_m} - \frac{1}{n-1} \sum_{r \neq i} (\Delta_{2 g_i g_j g_r} + \mu_{g_i g_j} \mu_{g_i g_r})
$$

$$
- \frac{1}{n-1} \sum_{r \neq i} (\Delta_{2 g_i g_m g_r} + \mu_{g_i g_m} \mu_{g_i g_r}) + \frac{1}{(n-1)^2} \sum_{r_1, r_2 \neq i} (\Delta_{2 g_i g_{r_1} g_{r_2}} + \mu_{g_i g_{r_1}} \mu_{g_i g_{r_2}}) \Big\}
$$

$$
\to \Delta_{i,2} := \frac{1}{n} \sum_{k \in \{1,2\}} \gamma_k \Big( \sum_{l \in \{1,2\}} \gamma_l \Delta_{2 g_i kl} + \sum_{l \in \{1,2\}} \gamma_l \mu_{g_i k} \mu_{g_i l} - \Delta_{2 g_i kk} - \mu_{g_i k}^2 \Big).
$$

$\square$

## A.3. Proof of Theorem 3.2

*Proof.* According to the definition of $Q_i$, we can rewrite $Q_i$ as $Q_i = \frac{1}{\sqrt{n}} \sum_{j=1}^n V_{ij}$, where $V_{ij}$ is defined as follows,

$$
V_{ij} = \begin{cases} \dfrac{\sqrt{n}\big[h(X_i, X_j) - \mu\big]}{(n_{g_i}-1)\Big[\big(\frac{1}{n_{g_i}-1} + \frac{1}{n-n_{g_i}}\big)(\Delta_1 - \Delta_2)\Big]^{1/2}}, & \text{if } j \in \mathcal{C}_{g_i}, \\[4mm] \dfrac{-\sqrt{n}\big[h(X_i, X_j) - \mu\big]}{(n-n_{g_i})\Big[\big(\frac{1}{n_{g_i}-1} + \frac{1}{n-n_{g_i}}\big)(\Delta_1 - \Delta_2)\Big]^{1/2}}, & \text{if } j \notin \mathcal{C}_{g_i}. \end{cases}
$$

Let $y_j = (y_{1j}, \cdots, y_{nj})^\top \in \mathbb{R}^n$ $(j = 1, \cdots, n)$ be independent Gaussian random vectors with mean zero and the same covariance matrix as $(Q_1, \cdots, Q_n)^\top$, i.e., $\Sigma$. Denote $Y = (Y_1, \cdots, Y_n)^\top$, where $Y_i = \frac{1}{\sqrt{n}} \sum_{j=1}^n y_{ij}$. In addition, define $Q^* = \max_{1 \leq i \leq n} Q_i$ and $Y^* = \max_{1 \leq i \leq n} Y_i$. To facilitate the proof of Theorem 3.2, we further define $T^* = \max_{1 \leq i \leq n} T_i$ and $Z^* = \max_{1 \leq i \leq n} Z_i$, where $T_i$ is the estimator of $Q_i$, $Z_i$ is the $i$-th component of $Z$, and $Z = (Z_1, \cdots, Z_n)^\top$ is a multivariate normal random vector with mean zero and covariance matrix $\Sigma$.

The proof of Theorem 3.2 consists of three steps. The first two steps demonstrate that, as $n \to \infty$,

$$
\sup_{t \in \mathbb{R}} |P(Q^* \leq t) - P(Y^* \leq t)| \to 0. \tag{4}
$$

The last step shows $Q^*$ and $Y^*$ in (4) can be replaced by $T^*$ and $Z^*$, respectively.

**Step 1.** Since the maximum function is non-differentiable, a smooth approximation of the maximum function is applied. For any vector $\boldsymbol{x} = (x_1, \cdots, x_n)^\top$, consider the smooth max function $F_\zeta(\boldsymbol{x}) = \zeta^{-1}\log(\sum_{i=1}^n e^{\zeta x_i})$, where $\zeta > 0$. Since $e^{\max_{1\le i\le n} \zeta x_i} \le \sum_{i=1}^n e^{\zeta x_i} \le ne^{\max_{1\le i\le n} \zeta x_i}$, we have $\max_{1\le i\le n} x_i \le F_\zeta(\boldsymbol{x}) \le \max_{1\le i\le n} x_i + \zeta^{-1}\log n$. Hence, we can obtain

$$P(Q^* \le t) = P\big\{\max_{1\le i\le n} Q_i \le t\big\} \le P\big\{F_\zeta(Q) \le t + \zeta^{-1}\log n\big\}. \tag{5}$$

For any $\zeta > 0$, denote $e_\zeta := \zeta^{-1}\log n$. Consider a function $g_0 \in C_b^3(\mathbb{R})$ mapped from $\mathbb{R}$ to $[0,1]$, where $C_b^3(\mathbb{R})$ is the set of functions having third continuous and bounded derivatives. Specifically, $g_0(s) = 1$ if $s \le 0$, $g_0(s) = 0$ if $s \ge 1$, and $g_0(s) \in [0,1]$ if $0 < s < 1$. Given $t \in \mathbb{R}$, define $g(s) = g_0(\psi(s - t - e_\zeta))$, where $\psi = n^{1/8}$ and $\zeta = \psi\log n$. In addition, define $G_k = \sup_{s\in\mathbb{R}} |\partial^k g(s)/\partial s|$. It can be verified that $G_0 = 1$, $G_1 \lesssim \psi$, $G_2 \lesssim \psi^2$ and $G_3 \lesssim \psi^3$. Furthermore, we have that $g(F_\zeta(Q)) = 1$ if $F_\zeta(Q) \le t + e_\zeta$ and $g(F_\zeta(Q)) \ge 0$ if $F_\zeta(Q) \ge t + e_\zeta$. These results lead to

$$P\big\{F_\zeta(Q) \le t + e_\zeta\big\} \le E\big\{g(F_\zeta(Q))\big\}. \tag{6}$$

This, together with (5), implies that

$$P(Q^* \le t) \le E\big\{g(F_\zeta(Q))\big\}. \tag{7}$$

**Step 2.** To show (4), this step mainly derives the bound of $E\{g(F_\zeta(Q))\} - E\{g(F_\zeta(Y))\}$. To this end, for any $t \in [0,1]$, define the Slepian interpolation between $Q$ and $Y$ as $H(t) = \sqrt{t}Q + \sqrt{1-t}Y = \sum_{j=1}^n H_j(t)$, where $H_j(t) = (H_{1j}(t), \cdots, H_{nj}(t))^\top \in \mathbb{R}^n$ and $H_{ij}(t) = \frac{1}{\sqrt{n}}(\sqrt{t}V_{ij} + \sqrt{1-t}y_{ij})$. Let $m(t) := g \circ F_\zeta$ and $\Psi(t) = E\{m(H(t))\}$. Then we have

$$E\big\{g(F_\zeta(Q))\big\} - E\big\{g(F_\zeta(Y))\big\}$$
$$= \Psi(1) - \Psi(0) = \int_0^1 \Psi'(t)dt$$
$$= \frac{1}{2}\sum_{i=1}^n \sum_{j=1}^n \int_0^1 E\Big\{\partial_i m\big(H(t)\big)\dot{H}_{ij}(t)\Big\}dt,$$

where $\dot{H}_{ij}(t) = \frac{1}{\sqrt{n}}\Big(\frac{1}{\sqrt{t}}V_{ij} - \frac{1}{\sqrt{1-t}}y_{ij}\Big)$. In addition, define Stein's leave-one-out version of $H(t)$ as

$$H^{(ij)}(t) := (h_1^{(ij)}(t), \cdots, h_n^{(ij)}(t))^\top = H(t) - H_i(t) - H_j(t), \ (1 \le i,j \le n).$$

For any $1 \le i \le n$, we employ Taylor's theorem and obtain that

$$\partial_i m\big(H(t)\big) = \partial_i m\big(H^{(ij)}(t) + H_i(t) + H_j(t)\big)$$
$$= \partial_i m\big(H^{(ij)}(t)\big) + \sum_{k=1}^n \partial_i \partial_k m\big(H^{(ij)}(t)\big)\big[H_{ki}(t) + H_{kj}(t)\big]$$
$$+ \sum_{k,l=1}^n \int_0^1 (1-\xi)\partial_i \partial_k \partial_l m\big(H^{(ij)}(t) + \xi H_i(t) + \xi H_j(t)\big)$$
$$\times [H_{ki}(t)H_{lj}(t) + H_{ki}(t)H_{li}(t) + H_{kj}(t)H_{lj}(t) + H_{kj}(t)H_{li}(t)]d\xi.$$

Accordingly,

$$E\big\{g(F_\zeta(Q))\big\} - E\big\{g(F_\zeta(Y))\big\} = \frac{1}{2}(I + II + III),$$

where

$$I = \sum_{i=1}^n \sum_{j=1}^n \int_0^1 E\Big\{\partial_i m\big(H^{(ij)}(t)\big)\dot{H}_{ij}(t)\Big\}dt,$$

$$II = \sum_{i=1}^n \sum_{j=1}^n \sum_{k=1}^n \int_0^1 E\big\{\partial_i \partial_k m\big(H^{(ij)}(t)\big)\dot{H}_{ij}(t)\big[H_{ki}(t) + H_{kj}(t)\big]\big\}dt, \text{ and}$$

$$III = \sum_{i=1}^n \sum_{j=1}^n \sum_{k=1}^n \sum_{l=1}^n \int_0^1 \int_0^1 (1-\xi)E\Big\{\partial_i \partial_k \partial_l m\big(H^{(ij)}(t) + \xi H_i(t) + \xi H_j(t)\big)\dot{H}_{ij}(t)$$

$$\times \big[ H_{ki}(t)H_{lj}(t) + H_{ki}(t)H_{li}(t) + H_{kj}(t)H_{lj}(t) + H_{kj}(t)H_{li}(t) \big] \Big\} d\xi dt.$$

We next show the bounds of $I, II$ and $III$ separately.

**Bound on $I$.** Define $e_r^{(ij)}(t) = (0, \cdots, 0, h_r^{(ij)}(t), 0, \cdots, 0)^\top \in \mathbb{R}^n$ and $\tilde{H}^{(ij)}(t) = H^{(ij)}(t) - e_i^{(ij)}(t) - e_j^{(ij)}(t)$. Then, by Taylor's theorem, we have

$$\partial_i m\big(H^{(ij)}(t)\big) = \partial_i m\big(\tilde{H}^{(ij)}(t)\big) + \sum_{k \in \{i,j\}} \partial_i \partial_k m\big(\tilde{H}^{(ij)}(t)\big) h_k^{(ij)}(t)$$

$$+ \sum_{k,l \in \{i,j\}} \int_0^1 (1-\xi) \partial_i \partial_k \partial_l m\big(\tilde{H}^{(ij)}(t) + \xi e_i^{(ij)}(t) + \xi e_j^{(ij)}(t)\big) h_k^{(ij)}(t) h_l^{(ij)}(t) d\xi.$$

As a result,

$$I = \sum_{i=1}^n \sum_{j=1}^n \int_0^1 E\left\{ \partial_i m\big(H^{(ij)}(t)\big) \dot{H}_{ij}(t) \right\} dt = I_1 + I_2 + I_3,$$

where

$$I_1 = \sum_{i=1}^n \sum_{j=1}^n \int_0^1 E\left\{ \partial_i m\big(\tilde{H}^{(ij)}(t)\big) \dot{H}_{ij}(t) \right\} dt,$$

$$I_2 = \sum_{i=1}^n \sum_{j=1}^n \sum_{k \in \{i,j\}} \int_0^1 E\left\{ \partial_i \partial_k m\big(\tilde{H}^{(ij)}(t)\big) h_k^{(ij)}(t) \dot{H}_{ij}(t) \right\} dt, \text{ and}$$

$$I_3 = \sum_{i=1}^n \sum_{j=1}^n \sum_{k,l \in \{i,j\}} \int_0^1 \int_0^1 (1-\xi) E\left\{ \partial_i \partial_k \partial_l m\big(\tilde{H}^{(ij)}(t) + \xi e_i^{(ij)}(t) + \xi e_j^{(ij)}(t)\big) h_k^{(ij)}(t) h_l^{(ij)}(t) \dot{H}_{ij}(t) \right\} d\xi dt.$$

We subsequently evaluate the terms $I_1, I_2$ and $I_3$. By the independence of $\tilde{H}^{(ij)}(t)$ and $\dot{H}_{ij}(t)$, together with the fact that $E[\dot{H}_{ij}(t)] = 0$, we have $I_1 = 0$.

Define $\omega(t) = 1/\min(\sqrt{t}, \sqrt{1-t})$. For the term $I_2$, we have

$$|I_2| \le (G_2 + 2G_1\zeta) \cdot \int_0^1 E\Big[ \max_{1 \le i,j \le n, k \in \{i,j\}} \big| h_k^{(ij)}(t) \dot{H}_{ij}(t) \big| \Big] dt$$

$$\le (G_2 + 2G_1\zeta) \cdot \int_0^1 \omega(t) \Big\{ E[\max_{i,j,k} |h_k^{(ij)}(t)|]^2 \cdot E[\max_{i,j} |\dot{H}_{ij}(t)/\omega(t)|]^2 \Big\}^{1/2} dt,$$

where the first inequality is by Lemma A.2, and the second inequality is by Hölder's inequality. Based on the construction of $h_k^{(ij)}(t)$, we can easily obtain that $E[\max_{i,j,k} |h_k^{(ij)}(t)|]^2 = O(\log n)$. This leads to

$$|I_2| \le (G_2 + 2G_1\zeta) \sqrt{\frac{\log n}{n}} \bar{E}\Big\{ \big[ \max_{1 \le j \le n} (|V_{ij}| + |y_{ij}|)^2 \big] \Big\}^{1/2}.$$

We lastly derive the bound of $|I_3|$. After algebraic simplification, we have that

$$|I_3| \lesssim \sum_{i=1}^n \sum_{j=1}^n \sum_{k,l \in \{i,j\}} \int_0^1 E\left\{ \partial_i \partial_k \partial_l \big(\tilde{H}^{(ij)}(t)\big) h_k^{(ij)}(t) h_l^{(ij)}(t) \dot{H}_{ij}(t) \right\} dt$$

$$\lesssim n^{-1}(G_3 + G_2\zeta + G_1\zeta^2) \int_0^1 E\Big[ \max_{1 \le i,j \le n, k,l \in \{i,j\}} \big| h_k^{(ij)}(t) h_l^{(ij)}(t) \dot{H}_{ij}(t) \big| \Big] dt$$

$$\lesssim n(G_3 + G_2\zeta + G_1\zeta^2) \int_0^1 E\Big[ \max_{1 \le i,k,l,j,r_1,r_2 \le n} \big| H_{kr_1}(t) H_{lr_2}(t) \dot{H}_{ij}(t) \big| \Big] dt,$$

where the first inequality is by Lemma A.3, the second inequality is by Lemma A.2, and the third inequality is due to the

definition of $h_k^{(ij)}(t)$. By Hölder's inequality, we have

$$\int_0^1 E\Big\{\max_{1\le i,k,l,j,r_1,r_2\le n}|\dot{H}_{ij}(t)H_{kr_1}(t)H_{lr_2}(t)|\Big\}dt$$

$$= \int_0^1 \omega(t)E\Big\{\max_{1\le i,k,l,j,r_1,r_2\le n}|\dot{H}_{ij}(t)/\omega(t)H_{kr_1}(t)H_{lr_2}(t)|\Big\}dt$$

$$\le \int_0^1 \omega(t)\Big\{E\big[\max_{i,j}|\dot{H}_{ij}(t)/\omega(t)|\big]^3 E\big[\max_{k,r_1}|H_{kr_1}(t)|\big]^3 E\big[\max_{l,r_2}|H_{lr_2}(t)|\big]^3\Big\}^{1/3}dt$$

$$\lesssim n^{-3/2}\bar{E}\big[\max_{1\le j\le n}(|V_{ij}|+|y_{ij}|)^3\big],$$

where the last inequality is due to the fact that $|\dot{H}_{ij}(t)/\omega(t)| \le (|V_{ij}|+|y_{ij}|)/\sqrt{n}$, $|H_{ij}(t)| \le (|V_{ij}|+|y_{ij}|)/\sqrt{n}$ and $\int_0^1 \omega(t)dt \lesssim 1$. Accordingly,

$$|I_3| \lesssim n^{-1/2}(G_3+G_2\zeta+G_1\zeta^2)\bar{E}\big[\max_{1\le j\le n}(|V_{ij}|+|y_{ij}|)^3\big].$$

This, together with the bounded results of $I_1$ and $I_2$, implies that

$$|I| \le n^{-1/2}(G_3+G_2\zeta+G_1\zeta^2)\bar{E}\big[\max_{1\le j\le n}(|V_{ij}|+|y_{ij}|)^3\big]+o(1). \tag{8}$$

**Bound on $II$.** The $II$ can be expressed as

$$II = \sum_{i=1}^n\sum_{j=1}^n\sum_{k=1}^n\int_0^1 E\Big\{\partial_i\partial_k m\big(H^{(ij)}(t)\big)\dot{H}_{ij}(t)H_{ki}(t)\Big\}dt$$

$$+ \sum_{i=1}^n\sum_{j=1}^n\sum_{k=1}^n\int_0^1 E\Big\{\partial_i\partial_k m\big(H^{(ij)}(t)\big)\dot{H}_{ij}(t)H_{kj}(t)\Big\}dt$$

$$=: II_1+II_2.$$

We next evaluate the bounds of $II_1$ and $II_2$. Since the technique for showing their bounds is similar, we only consider $II_2$. Define $H^{(ijk)}(t) = H^{(ij)}(t)-H_k(t) = (h_1^{(ijk)}(t),\cdots,h_n^{(ijk)}(t))^\top$. By Taylor's theorem, we obtain that

$$II_2 = \sum_{i=1}^n\sum_{j=1}^n\sum_{k=1}^n\int_0^1 E\Big\{\partial_i\partial_k m\big(H^{(ijk)}(t)+H_k(t)\big)\dot{H}_{ij}(t)H_{kj}(t)\Big\}dt$$

$$= \sum_{i=1}^n\sum_{j=1}^n\sum_{k=1}^n\int_0^1 E\Big\{\partial_i\partial_k m\big(H^{(ijk)}(t)\big)\dot{H}_{ij}(t)H_{kj}(t)\Big\}dt$$

$$+ \sum_{i=1}^n\sum_{j=1}^n\sum_{k=1}^n\sum_{l=1}^n\int_0^1\int_0^1 (1-\xi)E\Big[\partial_i\partial_k\partial_l m\big(H^{(ijk)}(t)+\xi H_k(t)\big)\times \dot{H}_{ij}(t)H_{kj}(t)H_{lk}(t)\Big]d\xi dt$$

$$=: \Omega_1+\Omega_2.$$

By Lemmas A.2 and A.3, we have

$$|\Omega_2| \lesssim \sum_{i=1}^n\sum_{j=1}^n\sum_{k=1}^n\sum_{l=1}^n\int_0^1\int_0^1 (1-\xi)E\Big[\partial_i\partial_k\partial_l m\big(H^{(ijk)}(t)\big)\dot{H}_{ij}(t)H_{kj}(t)H_{lk}(t)\Big]d\xi dt$$

$$\le (G_3+G_2\zeta+G_1\zeta^2)\sum_{j=1}^n\int_0^1 E\Big\{\max_{1\le i,k,l\le n}|\dot{H}_{ij}(t)H_{kj}(t)H_{lk}(t)|\Big\}dt$$

$$\lesssim n(G_3+G_2\zeta+G_1\zeta^2)\int_0^1 E\Big\{\max_{1\le i,j,k,l\le n}|\dot{H}_{ij}(t)H_{kj}(t)H_{lk}(t)|\Big\}dt$$

$$\lesssim n^{-1/2}(G_3+G_2\zeta+G_1\zeta^2)\bar{E}\big[\max_{1\le j\le n}(|V_{ij}|+|y_{ij}|)^3\big].$$

Define $e_r^{(ijk)}(t) = (0, \cdots, 0, h_r^{(ijk)}(t), 0, \cdots, 0)^\top \in \mathbb{R}^n$ and $\tilde{H}^{(ijk)}(t) = H^{(ijk)}(t) - e_i^{(ijk)}(t) - e_j^{(ijk)}(t) - e_k^{(ijk)}(t)$. Employing Taylor's theorem, we obtain that

$$
\Omega_1 = \sum_{i=1}^n \sum_{j=1}^n \sum_{k=1}^n \int_0^1 E\left\{ \partial_i \partial_k m\big(\tilde{H}^{(ijk)}(t)\big) \dot{H}_{ij}(t) H_{kj}(t) \right\} dt
$$

$$
+ \sum_{i=1}^n \sum_{j=1}^n \sum_{k=1}^n \sum_{l \in \{i,j,k\}} \int_0^1 \int_0^1 (1-\xi) E\Big\{ \partial_i \partial_k \partial_l m\big[\tilde{H}^{(ijk)}(t) + \xi e_i^{(ijk)}(t) + \xi e_j^{(ijk)}(t) + \xi e_k^{(ijk)}(t)\big]
$$

$$
\times \dot{H}_{ij}(t) H_{kj}(t) h_l^{(ijk)}(t) \Big\} d\xi dt
$$

$$
= \Omega_{1,1} + \sum_{u=2}^4 \Omega_{1,u}.
$$

By the definitions of $\dot{H}_{ij}(t)$ and $H_{kj}(t)$ in the beginning of Step 2, we have $E[\dot{H}_{ij}(t) H_{kj}(t)] = n^{-1} E[V_{ij} V_{kj} - y_{ij} y_{kj}] = 0$. Since $\tilde{H}^{(ijk)}(t)$ is independent of $\dot{H}_{ij}(t)$ and $H_{kj}(t)$, we obtain that $\Omega_{1,1} = 0$. We then show the bounds of $\Omega_{1,u}$ for $u = 2, \cdots, 4$. By Lemma A.2, we have

$$
|\Omega_{1,u}| \lesssim \sum_{i=1}^n \sum_{j=1}^n \sum_{k=1}^n \int_0^1 E\Big\{ \partial_i \partial_k \partial_l m\big[\tilde{H}^{(ijk)}(t)\big] \dot{H}_{ij}(t) H_{kj}(t) h_l^{(ijk)}(t) \Big\} d\xi dt
$$

$$
\leq (G_3 + G_2 \zeta + G_1 \zeta^2) \int_0^1 E\Big\{ \max_{1 \leq i,j,k \leq n} |\dot{H}_{ij}(t) H_{kj}(t) h_l^{(ijk)}(t)| \Big\} dt
$$

$$
\lesssim n(G_3 + G_2 \zeta + G_1 \zeta^2) \int_0^1 E\Big\{ \max_{1 \leq i,j,k,l,s \leq n} |\dot{H}_{ij}(t) H_{kj}(t) H_{ls}(t)| \Big\} dt
$$

$$
\leq n(G_3 + G_2 \zeta + G_1 \zeta^2) \int_0^1 \omega(t) \Big\{ E\big[ \max_{1 \leq i,j \leq n} |\dot{H}_{ij}(t)/\omega(t)|^3 \big]
$$

$$
\times E\big[ \max_{1 \leq j,k \leq n} |H_{kj}(t)|^3 \big] E\big[ \max_{1 \leq l,s \leq n} |H_{ls}(t)|^3 \big] \Big\}^{1/3} dt
$$

$$
\lesssim n^{-1/2}(G_3 + G_2 \zeta + G_1 \zeta^2) \bar{E}\big[ \max_{1 \leq j \leq n} (|V_{ij}| + |y_{ij}|)^3 \big],
$$

where the fourth inequality is by Hölder's inequality. This, in conjunction with above results, leads to

$$
|II_2| \lesssim n^{-1/2}(G_3 + G_2 \zeta + G_1 \zeta^2) \bar{E}\big[ \max_{1 \leq j \leq n} (|V_{ij}| + |y_{ij}|)^3 \big]. \tag{9}
$$

**Bound on $III$.** To bound $III$, we separate it into four components. The first component is defined below.

$$
III_1 = \sum_{i=1}^n \sum_{j=1}^n \sum_{k=1}^n \sum_{l=1}^n \int_0^1 \int_0^1 (1-\xi) E\Big\{ \partial_i \partial_k \partial_l m\big(H^{(ij)}(t)
$$

$$
+ \xi H_i(t) + \xi H_j(t)\big) \times \dot{H}_{ij}(t) H_{ki}(t) H_{lj}(t) \Big\} d\xi dt.
$$

Analogously, $III_2$, $III_3$ and $III_4$ can also be defined. By Lemma A.2, we have

$$
|III_1| \leq (G_3 + G_2 \zeta + G_1 \zeta^2) \sum_{j=1}^n \int_0^1 E\Big[ \max_{1 \leq i,k,l \leq n} \dot{H}_{ij}(t) H_{ki}(t) H_{lj}(t) \Big] dt
$$

$$
\lesssim n(G_3 + G_2 \zeta + G_1 \zeta^2) \int_0^1 E\Big[ \max_{1 \leq i,j,k,l \leq n} \dot{H}_{ij}(t) H_{ki}(t) H_{lj}(t) \Big] dt
$$

$$
\lesssim n^{-1/2}(G_3 + G_2 \zeta + G_1 \zeta^2) \bar{E}\big[ \max_{1 \leq j \leq n} (|V_{ij}| + |y_{ij}|)^3 \big].
$$

Similar techniques can be applied to $III_2, III_3$ and $III_4$. Accordingly,

$$
|III| \lesssim n^{-1/2}(G_3 + G_2 \zeta + G_1 \zeta^2) \bar{E}\big[ \max_{1 \leq j \leq n} (|V_{ij}| + |y_{ij}|)^3 \big]. \tag{10}
$$

This, together with (8) and (9), yields

$$\left| E\{g(F_\zeta(Q))\} - E\{g(F_\zeta(Y))\} \right| \leq n^{-1/2}(G_3 + G_2\zeta + G_1\zeta^2)\bar{E}\left[ \max_{1 \leq j \leq n} (|V_{ij}| + |y_{ij}|)^3 \right] + o(1). \tag{11}$$

Under Condition (C1), all $V_{ij}$s have the same order of $O(1)$. Applying Bonferroni's inequality, we then have that

$$\bar{E}\left[ \max_{1 \leq j \leq n} (|V_{ij}| + |y_{ij}|)^3 \right] = O(\log^{3/2} n).$$

As a result,

$$\left| E\{g(F_\zeta(Q))\} - E\{g(F_\zeta(Y))\} \right| \leq n^{-1/2}(G_3 + G_2\zeta + G_1\zeta^2)\log^{3/2} n + o(1), \tag{12}$$

which yields the bound of $E\{g(F_\zeta(Q))\} - E\{g(F_\zeta(Y))\}$.

The results of (7) and (12) lead to

$$\begin{aligned}
P(Q^* \leq t) &\leq E\{g(F_\zeta(Q))\} \\
&\leq E\{g(F_\zeta(Y))\} + Cn^{-1/2}\log^{3/2} n(G_3 + G_2\psi + G_1\psi^2) + o(1) \\
&\leq P(F_\zeta(Y) \leq t + e_\zeta + \psi^{-1}) + Cn^{-1/2}\log^{3/2} n(G_3 + G_2\psi + G_1\psi^2) + o(1) \\
&\leq P(Y^* \leq t + e_\zeta + \psi^{-1}) + Cn^{-1/2}\log^{3/2} n(G_3 + G_2\psi + G_1\psi^2) + o(1),
\end{aligned}$$

where the third inequality is due to the construction of $g$, and the last inequality is based on the fact that $\max_i x_i \leq F_\zeta(\boldsymbol{x})$. By Lemma A.5, we have

$$P(Y^* \leq t + e_\zeta + \psi^{-1}) \leq P(Y^* \leq t) + C(e_\zeta + \psi^{-1})\sqrt{\log(n\psi)}.$$

This implies that

$$P(Q^* \leq t) \leq P(Y^* \leq t) + o(1).$$

Applying techniques similar to those used above, we can show that

$$P(Q^* \leq t) \geq P(Y^* \leq t) + o(1).$$

which completes the entire proof of (4).

**Step 3.** Under the null hypothesis of $H_0$, we apply the Bonferroni and Bernstein inequalities and obtain that $\max_{1 \leq i \leq n} |\hat{\Delta}_{i,1} - \Delta_1| = O_p(\sqrt{\log n/n})$ and $\max_{1 \leq i \leq n} |\hat{\Delta}_{i,2} - \Delta_2| = O_p(\sqrt{\log n/n})$. In addition, by the result of (4), we have $\max_{1 \leq i \leq n} |Q_i| = O_p(\sqrt{\log n})$. The above results lead to

$$\begin{aligned}
&\left| \max_{1 \leq i \leq n} Q_i - \max_{1 \leq i \leq n} T_i \right| \\
&= \left| \max_{1 \leq i \leq n} \frac{\hat{d}_i^{(in)} - \hat{d}_i^{(bet)}}{\left\{ \left( \frac{1}{n_{g_i}-1} + \frac{1}{n-n_{g_i}} \right)(\Delta_1 - \Delta_2) \right\}^{1/2}} \right. \\
&\quad \left. - \max_{1 \leq i \leq n} \frac{\hat{d}_i^{(in)} - \hat{d}_i^{(bet)}}{\left\{ \left( \frac{1}{n_{g_i}-1} + \frac{1}{n-n_{g_i}} \right)(\hat{\Delta}_{i,1} - \hat{\Delta}_{i,2}) \right\}^{1/2}} \right| \\
&\leq C \max_{1 \leq i \leq n} |Q_i| \max_{1 \leq i \leq n} |\Delta_1 - \hat{\Delta}_{i,1} - \Delta_2 + \hat{\Delta}_{i,2}| \\
&= O_p(\log n/\sqrt{n}) \to 0.
\end{aligned}$$

This implies that, for any $t \in \mathbb{R}$,

$$P(\max_{1 \leq i \leq n} Q_i \leq t) - P(\max_{1 \leq i \leq n} T_i \leq t) \to 0, \tag{13}$$

as $n \to \infty$.

We next prove that, for any $t \in \mathbb{R}$,

$$P(\max_{1 \leq i \leq n} Y_i \leq t) - P(\max_{1 \leq i \leq n} Z_i \leq t) \to 0, \tag{14}$$

as $n \to \infty$. To this end, define $\hat{\sigma}_{ij}$ as the covariance of $Y_i$ and $Y_j$, and define $\sigma_{ij}$ as the covariance of $Z_i$ and $Z_j$. Under

the null hypothesis $H_0$, we have $\max_{1 \leq i,j \leq n} |\hat{\sigma}_{ij} - \sigma_{ij}| = O_p(n^{-1/2})$ by the Bernstein inequality. In addition, set $\Delta_0 = O_p(n^{-1/2})$, $C_1 = 2\lambda_{\max}(\Sigma)$, and $c_1 = \lambda_{\min}(\Sigma)/2$ in Lemma A.4. This setting, in conjunction with the above result and Lemma A.4, implies (14).

Combining the results in (4), (13) and (14), we obtain that

$$P(T^* \leq t) - P(Z^* \leq t) \to 0. \tag{15}$$

By following the same procedure as in Chernozhukov et al. (2017), we can demonstrate that the above Gaussian approximation holds in all hyper-rectangles of $\mathbb{R}^n$. Considering the special hyper-rectangles $[-t, t]^n$, we then obtain that

$$P(\max_{1 \leq i \leq n} |T_i| \leq t) - P(\max_{1 \leq i \leq n} |Z_i| \leq t) \to 0, \tag{16}$$

which completes the entire proof. $\square$

## A.4. Proof of Theorem 3.4

*Proof.* To prove this theorem, we consider the following two steps.

**Step 1.** Note that $Z = (Z_1, \cdots, Z_n)^\top \in \mathbb{R}^n$ is a multivariate normal random vector with mean zero and covariance matrix $\Sigma$. Hence, for any $i = 1, \cdots, n$, $Z_i$ follows a normal distribution with mean 0 and finite variance. Moreover, by the Bonferroni inequality, we have

$$P\left\{ \max_{1 \leq i \leq n} Z_i^2 > (2 + \epsilon/3) \log n \right\} \leq \sum_{i=1}^n P\left\{ Z_i^2 > (2 + \epsilon/3) \log n \right\} \to 0.$$

To prove the consistency of the proposed test, the above result and (16) together indicate that it will suffice to show that $P(\max_{1 \leq i \leq n} T_i^2 > (2 + \epsilon/3) \log n) \to 1$ under the alternative hypothesis $H_1$.

**Step 2.** Define

$$\bar{Q}_i = \frac{\hat{d}_i^{(in)} - \hat{d}_i^{(bet)} - (\mu_{g_i}^{(in)} - \mu_{g_i}^{(bet)})}{\left\{ \left( \frac{1}{n_{g_i} - 1} + \frac{1}{n - n_{g_i}} \right) (\Delta_{i,1} - \Delta_{i,2}) \right\}^{1/2}},$$

and then denote $(\bar{Q}_1, \cdots, \bar{Q}_n)^\top$. Following techniques similar to those used in the proof of Theorem 3.2, we obtain that

$$P(\max_{1 \leq i \leq n} \bar{Q}_i < x) - P(\max_{1 \leq i \leq n} U_i < x) \to 0, \tag{17}$$

where $U_i$ is the $i$-th component of $U$, and $U = (U_1, \cdots, U_n)^\top \in \mathbb{R}^n$ is a multivariate normal random vector with mean zero and the same covariance matrix as $(\bar{Q}_1, \cdots, \bar{Q}_n)^\top$. Accordingly, for any $i = 1, \cdots, n$, $U_i$ is a normal distribution with mean 0 and finite variance. In addition, by the Bonferroni inequality, we have

$$P\left\{ \max_{1 \leq i \leq n} U_i^2 > (2 + \epsilon/3) \log n \right\} \leq \sum_{i=1}^n P\left\{ U_i^2 > (2 + \epsilon/3) \log n \right\} \to 0.$$

This, in conjunction with (17), implies that

$$P\left\{ \max_{1 \leq i \leq n} \bar{Q}_i^2 > (2 + \epsilon/3) \log n \right\} \to 0. \tag{18}$$

Hence,

$$\max_{1 \leq i \leq n} |\bar{Q}_i| = O_p(\sqrt{\log n}). \tag{19}$$

Under the alternative hypothesis $H_1$, we employ the Bonferroni and Bernstein inequalities and obtain that

$$\max_{1 \leq i \leq n} |\hat{\Delta}_{i,1} - \Delta_{i,1}| = O_p(\sqrt{\log n / n}) \text{ and}$$

$$\max_{1 \leq i \leq n} |\hat{\Delta}_{i,2} - \Delta_{i,2}| = O_p(\sqrt{\log n / n}). \tag{20}$$

Define

$$\mathring{T}_i = \frac{\hat{d}_i^{(in)} - \hat{d}_i^{(bet)} - (\mu_{g_i}^{(in)} - \mu_{g_i}^{(bet)})}{\left\{ \left( \frac{1}{n_{g_i}-1} + \frac{1}{n-n_{g_i}} \right) (\hat{\Delta}_{i,1} - \hat{\Delta}_{i,2}) \right\}^{1/2}},$$

and let $\mathring{T} = (\mathring{T}_1, \cdots, \mathring{T}_n)^\top$. By (20), we have that $|\max_{1\le i\le n} \bar{Q}_i - \max_{1\le i\le n} \mathring{T}_i| = O_p(\log n/\sqrt{n}) \to 0$. This, together with (18), implies that

$$P\left\{ \max_{1\le i\le n} \mathring{T}_i^2 > (2+\epsilon/3)\log n \right\} \to 0. \tag{21}$$

By the triangle inequality, the assumption in Theorem 3.4, (20), and (21), we have that, as $n \to \infty$,

$$\max_{1\le i\le n} T_i^2 \ge \max_{1\le i\le n} \frac{(\mu_{g_i}^{(in)} - \mu_{g_i}^{(bet)})^2}{\left( \frac{1}{n_{g_i}-1} + \frac{1}{n-n_{g_i}} \right) (\hat{\Delta}_{i,1} - \hat{\Delta}_{i,2})} - \max_{1\le i\le n} \mathring{T}_i^2$$

$$\ge \max_{1\le i\le n} \nu_i - \max_{1\le i\le n} \mathring{T}_i^2 - C\sqrt{\frac{\log n}{n}} \max_{1\le i\le n} \nu_i$$

$$\ge (4+\epsilon)\log n - (2+\epsilon/3)\log n - C\sqrt{\frac{\log n}{n}} \max_{1\le i\le n} \nu_i,$$

with probability tending to one. Accordingly, if $\max_{1\le i\le n} \nu_i = O(\log n)$, then

$$P\left\{ \max_{1\le i\le n} T_i^2 > (2+\epsilon/3)\log n \right\} \ge P\left\{ \max_{1\le i\le n} T_i^2 > (2+2\epsilon/3)\log n - O(\sqrt{(\log n)^3/n}) \right\} \to 1.$$

Suppose that $(\log n)^{-1} \max_{1\le i\le n} \nu_i \to \infty$, we then employ Theorem 3.4's assumption and (19), and obtain that there exists a constant $C$ such that, as $n \to \infty$,

$$\max_{1\le i\le n} T_i^2 \ge \max_{1\le i\le n} \frac{C(\hat{\mu}_i^{(in)} - \hat{\mu}_i^{(bet)})^2}{\left( \frac{1}{n_{g_i}-1} + \frac{1}{n-n_{g_i}} \right) (\Delta_{i,1} - \Delta_{i,2})}$$

$$\ge \max_{1\le i\le n} C\nu_i - \max_{1\le i\le n} C\bar{Q}_i^2$$

$$> (2+\epsilon/3)\log n,$$

with probability tending to one. This completes the entire proof.

$\square$

## B. MOD test for K-sample Problem

To establish the limiting null distribution of the proposed test statistic $T_\tau$, we need to find the covariance matrix of $(T_{\tau,1}, \cdots, T_{\tau,n})^\top$. To this end, we define

$$Q_{\tau,i} = \frac{\hat{d}_{\tau,i}^{(in)} - \hat{d}_{\tau,i}^{(bet)}}{\left\{ \left( \frac{1}{n_{g_i}-1} + \frac{1}{n-n_{g_i}} \right) (\Delta_1 - \Delta_2) \right\}^{1/2}} = \alpha(g_i)\big(\hat{d}_{\tau,i}^{(in)} - \hat{d}_{\tau,i}^{(bet)}\big),$$

where $\alpha(g_i) = \left\{ \left( \frac{1}{n_{g_i}-1} + \frac{1}{n-n_{g_i}} \right) (\Delta_1 - \Delta_2) \right\}^{-1/2}$. Let $\Sigma_\tau = (\sigma_{\tau,ij}) \in \mathbb{R}^{n\times n}$ be the covariance matrix of $(Q_{\tau,1}, \cdots, Q_{\tau,n})^\top$. Under $H_0$, the explicit expression of the covariance matrix $\Sigma_\tau$ is shown as

$$\sigma_{\tau,ij} = \begin{cases} \alpha^2(g_i)\left[ \frac{\Delta_1 + (n_{g_i}-4)\Delta_2}{(n_{g_i}-1)^2} + \frac{\Delta_2}{n-n_{g_i}} \right], & \text{if } g_i = g_j, \\ \alpha(g_i)\alpha(g_j)\frac{\Delta_1 - (n+2)\Delta_2}{(n-n_{g_i})(n-n_{g_j})}, & \text{if } g_i \ne g_j. \end{cases}$$

After carefully investigating the relationship between $Q_{\tau,i}$s and $T_{\tau,i}$s, the asymptotic distribution of $T_\tau$ is shown in Theorem 5.1.

To investigate the consistency of the test, we will study the behavior of $\hat{\Delta}_{\tau,i}^{(1)}$ and $\hat{\Delta}_{\tau,i}^{(2)}$ involved in $T_{\tau,i}$ under the alternative hypothesis $H_1$. Following the proof of Proposition 3.3 in the Appendix, we can obtain that $E(\hat{\Delta}_{\tau,i}^{(1)})$ and $E(\hat{\Delta}_{\tau,i}^{(2)})$ converge

to $\Delta_{\tau,i}^{(1)}$ and $\Delta_{\tau,i}^{(2)}$, respectively. To this end, we define

$$\nu_{i,\tau} = \frac{(\mu_{g_i g_i} - \sum_{k \neq g_i} \frac{\gamma_k}{1 - \gamma_{g_i}} \mu_{g_i k})^2}{\left(\frac{1}{n_{g_i} - 1} + \frac{1}{n - n_{g_i}}\right)(\Delta_{\tau,i}^{(1)} - \Delta_{\tau,i}^{(2)})}.$$

Using similar techniques as those in Section 3.3, the consistency of the proposed test is established in Theorem 5.2.

## C. Additional experimental results

### C.1. Comparison of Computational Time

In this subsection, we compare the computational time of all methods across different sample sizes under the mean distribution shift setting in Experiment II. The results, reported in Table S.1, indicate that the proposed method remains feasible even for relatively large sample sizes.

*Table S.1.* Average computational time (in seconds) of all methods under the mean distribution shift in Experiment II.

| $n$ | MODboot | MOD-fuse | MMD-split | MMD-median | MMD-fuse | ME | MMD-agglnc | MMD-agg |
|------|---------|----------|-----------|------------|----------|--------|------------|---------|
| 100 | 0.0094 | 2.2892 | 0.0020 | 0.0025 | 0.0087 | 0.5906 | 0.0507 | 0.0583 |
| 300 | 0.0227 | 3.6578 | 0.0056 | 0.0094 | 0.0501 | 0.3380 | 0.0674 | 0.0858 |
| 500 | 0.0442 | 7.2929 | 0.0124 | 0.0296 | 0.1188 | 0.3687 | 0.1037 | 0.1238 |
| 1000 | 0.1187 | 26.5842 | 0.0568 | 0.0944 | 0.4236 | 0.3510 | 0.2117 | 0.1908 |
| 2000 | 0.5759 | 108.4148 | 0.2220 | 0.3629 | 1.5043 | 0.4211 | 0.2953 | 0.2630 |

### C.2. Sensitivity to the Choice of Kernel

To investigate the sensitivity of the proposed test to the choice of kernel, we conduct additional experiments using the Laplace kernel under the same setting as Experiment III. Since MMD-aggInc and ME do not support the Laplace kernel, these methods are excluded, and the results for the remaining methods are reported in Table S.2. The results show that the proposed method exhibits similar performance across different kernel choices, indicating that it is relatively insensitive to the choice of kernel.

*Table S.2.* Simulation results for Experiment III under the Laplace kernel.

| Type | $\delta$ | MODboot | MOD-fuse | MMD-median | MMD-split | MMD-fuse | MMD-agg |
|------|------|---------|----------|------------|-----------|----------|---------|
| Mean | 0.8 | 0.475 | 0.525 | 0.500 | 0.220 | 0.420 | 0.485 |
| | 0.9 | 0.565 | 0.670 | 0.620 | 0.270 | 0.530 | 0.625 |
| | 1 | 0.675 | 0.775 | 0.740 | 0.310 | 0.625 | 0.745 |
| | 1.1 | 0.755 | 0.825 | 0.835 | 0.350 | 0.685 | 0.855 |
| | 1.2 | 0.795 | 0.865 | 0.900 | 0.400 | 0.760 | 0.915 |
| Covariance | 0.8 | 0.550 | 0.585 | 0.135 | 0.085 | 0.180 | 0.145 |
| | 0.9 | 0.610 | 0.645 | 0.145 | 0.080 | 0.190 | 0.150 |
| | 1 | 0.660 | 0.680 | 0.165 | 0.080 | 0.205 | 0.165 |
| | 1.1 | 0.695 | 0.705 | 0.175 | 0.080 | 0.220 | 0.175 |
| | 1.2 | 0.745 | 0.755 | 0.200 | 0.085 | 0.220 | 0.200 |
| Location | 0.8 | 0.635 | 0.670 | 0.205 | 0.105 | 0.220 | 0.205 |
| | 0.9 | 0.695 | 0.725 | 0.230 | 0.125 | 0.225 | 0.250 |
| | 1 | 0.750 | 0.750 | 0.275 | 0.125 | 0.255 | 0.265 |
| | 1.1 | 0.790 | 0.790 | 0.300 | 0.135 | 0.285 | 0.290 |
| | 1.2 | 0.820 | 0.795 | 0.325 | 0.145 | 0.295 | 0.310 |

### C.3. Comparison of MOD and MMD

In this subsection, we conduct an additional ablation study comparing MOD and MMD without employing any kernel selection mechanism. Specifically, under Experiment II with $n = 250$ and the Gaussian kernel, we evaluate the empirical

power of MOD and MMD across a range of bandwidth choices. The bandwidths are selected as the $0.1, 0.3, 0.5, 0.7$, and $0.9$ quantiles of the pairwise distances computed from the pooled observations.

The results are reported in Table S.3. Overall, MOD demonstrates superior performance under covariance shift, location shift, and distribution shift alternatives, while being only slightly less powerful under mean shift alternatives. These findings are broadly consistent with those reported in the main text.

*Table S.3.* Power comparison for MOD and MMD under different choice of bandwidths.

| Type | quantile | MOD | MMD |
|------|----------|------|------|
| Mean | 0.1 | 0.790 | 0.955 |
|  | 0.3 | 0.775 | 0.965 |
|  | 0.5 | 0.785 | 0.965 |
|  | 0.7 | 0.760 | 0.975 |
|  | 0.9 | 0.725 | 0.980 |
| Covariance | 0.1 | 0.720 | 0.640 |
|  | 0.3 | 0.695 | 0.540 |
|  | 0.5 | 0.685 | 0.385 |
|  | 0.7 | 0.665 | 0.285 |
|  | 0.9 | 0.650 | 0.210 |
| Location | 0.1 | 0.560 | 0.565 |
|  | 0.3 | 0.580 | 0.440 |
|  | 0.5 | 0.570 | 0.395 |
|  | 0.7 | 0.545 | 0.370 |
|  | 0.9 | 0.510 | 0.305 |
| Distribution | 0.1 | 0.505 | 0.480 |
|  | 0.3 | 0.680 | 0.585 |
|  | 0.5 | 0.790 | 0.640 |
|  | 0.7 | 0.870 | 0.645 |
|  | 0.9 | 0.935 | 0.575 |

