# OpenReview forum: "Kernel-based  Maximum-of-difference Test for Two-sample Comparison"
_ICML.cc/2026/Conference — ICML 2026 regular_

### Official Review · Reviewer_pymn · 2026-02-14

**Soundness:** 3
**Presentation:** 4
**Significance:** 3
**Originality:** 4
**Overall Recommendation:** 5
**Confidence:** 3

**Summary:**

This work studies an important problem in nonparametric two-sample testing, namely the limitations of kernel-based maximum mean discrepancy (MMD) tests under certain alternatives such as covariance and location shifts.
This paper's core contribution is about the introduction of a kernel-based maximum-of-difference (MOD) statistic that replaces the global aggregation in MMD with a maximization over localized discrepancies between within-sample and between-sample kernel distances.
The authors provide a careful theoretical analysis of the proposed statistic, including asymptotic null distributions and consistency guarantees, and extend the framework to a fused multi-kernel version (MOD-FUSE) as well as a K-sample setting.
A sequence of experiments on synthetic benchmarks and MNIST demonstrate that the proposed methods are competitive.

**Compliance With Llm Reviewing Policy:**

Affirmed.

**Final Justification:**

This paper's core contribution pertains to a novel kernel-based maximum-of-difference (MOD) test that addresses a well-identified limitation of MMD --- namely, cancellation effects that reduce power under covariance-shift and location-shift alternatives. This work aims to study a pertinent problem that is both theoretically interesting and practically important across generative modeling, domain adaptation, and scientific hypothesis testing.
The key insight --- replacing the sum-of-differences aggregation in MMD with a maximum --- is elegant and well-motivated by the illustrative toy example and Figure 1. The theoretical contributions are solid: the asymptotic null distribution is rigorously derived using Gaussian approximation techniques, and consistency is established under mild conditions.

The rebuttal addressed my main concerns.

**Key Questions For Authors:**

How sensitive is the performance of MOD to the choice of kernel family beyond Gaussian kernels?
Have the authors explored other kernels or adaptive bandwidth strategies?

The consistency result depends on a condition involving the maximum of $\nu_i$.
Can the authors provide more intuition or empirical illustrations of how this condition relates to realistic effect sizes?

**Limitations:**

Yes.

**Strengths And Weaknesses:**

Strengths

The derivation of the asymptotic null distribution and consistency results is thorough.
The ability to approximate critical values numerically without heavy resampling is an appealing feature.

The fused MOD procedure and the K-sample extension significantly increase the applicability of the method.
The kernel fusion strategy is well integrated and avoids some of the pitfalls of scale mismatch.

Weaknesses

While the method is conceptually simple, the computational complexity of maximizing over all observations and estimating covariance structures may become substantial for very large datasets.

---

> ### Author Rebuttal · Authors · 2026-03-30
>
> **Q1: The computational complexity of maximizing over all observations and estimating covariance structures?**
>
> **A1:**  Thanks for the suggestion. Although the proposed method involves maximizing over all observations and estimating covariance-related quantities, its computational cost remains manageable. For each $i$, the quantities $\\hat\{\\mu\}\_i$,    $\\hat{\\Delta}\_{i,1}$,  and $\\hat{\\Delta}\_{i,2}$  can be computed in  $O(n)$  time via row-wise summation. Therefore, evaluating the statistic over all $i = 1, \\ldots, n$ requires $O(n^2)$ operations in total. The covariance estimation has the same order of complexity.
>
> To further demonstrate practical scalability, we compare the running times of all methods across different sample sizes under a setup similar to Experiment II. The results in Table A.1 show that the computation time of our method remains acceptable even for relatively large sample sizes.
>
> **Table A.1.** Average computational time (in seconds) of all methods under the mean distribution shift in Experiment II.
>
> | n | MODboot | MOD_fuse | MMD_split | MMD_median | MMD_fuse | ME | MMD_aggInc | MMD_agg |
> |---|---:|---:|---:|---:|---:|---:|---:|---:|
> | 100  | 0.0094 | 2.2892 | 0.0020 | 0.0025 | 0.0087 | 0.5906 | 0.0507 | 0.0583 |
> | 300  | 0.0227 | 3.6578 | 0.0056 | 0.0094 | 0.0501 | 0.3380 | 0.0674 | 0.0858 |
> | 500  | 0.0442 | 7.2929 | 0.0124 | 0.0296 | 0.1188 | 0.3687 | 0.1037 | 0.1238 |
> | 1000 | 0.1187 | 26.5842 | 0.0568 | 0.0944 | 0.4236 | 0.3510 | 0.2117 | 0.1908 |
> | 2000 | 0.5759 | 108.4148 | 0.2220 | 0.3629 | 1.5043 | 0.4211 | 0.2953 | 0.2630 |
>
> ---
>
> **Q2: Sensitivity of the choice of kernel family?**
>
> **A2:**  Following your suggestion, we conducted additional experiments using the Laplace kernel under the same setup as Experiment III. Since MMD-aggInc and ME do not support the Laplace kernel, we exclude them and report the results for the remaining methods in Table A.2. The results show that our method performs similarly across different kernel choices, suggesting that it is not particularly sensitive to the kernel family.
>
> **Table A.2.** Simulation results for Experiment III under the Laplace kernel.
>
> | Type | delta | MODboot | MOD_fuse | MMD_median | MMD_split | MMD_fuse | MMD_agg |
> |---|---:|---:|---:|---:|---:|---:|---:|
> | Mean | 0.8 | 0.475 | 0.525 | 0.500 | 0.220 | 0.420 | 0.485 |
> |  | 0.9 | 0.565 | 0.670 | 0.620 | 0.270 | 0.530 | 0.625 |
> || 1.0 | 0.675 | 0.775 | 0.740 | 0.310 | 0.625 | 0.745 |
> |  | 1.1 | 0.755 | 0.825 | 0.835 | 0.350 | 0.685 | 0.855 |
> | | 1.2 | 0.795 | 0.865 | 0.900 | 0.400 | 0.760 | 0.915 |
> | Covariance | 0.8 | 0.550 | 0.585 | 0.135 | 0.085 | 0.180 | 0.145 |
> |  | 0.9 | 0.610 | 0.645 | 0.145 | 0.080 | 0.190 | 0.150 |
> | | 1.0 | 0.660 | 0.680 | 0.165 | 0.080 | 0.205 | 0.165 |
> | | 1.1 | 0.695 | 0.705 | 0.175 | 0.080 | 0.220 | 0.175 |
> | | 1.2 | 0.745 | 0.755 | 0.200 | 0.085 | 0.220 | 0.200 |
>
> ---
>
> **Q3: More intuition or empirical illustrations of how the maximum of $\\nu_i$ relates to realistic effect sizes?**
>
> **A3:**  Thanks for the helpful comment. We provide intuition for the condition involving $\\max_{1 \\le i \\le n} \\nu_i$ from three aspects.
>
> First, $\\mu_{g_i}^{(\\mathrm{in})} - \\mu_{g_i}^{(\\mathrm{bet})}$ measures the discrepancy between the within-sample and between-sample distances for observation $i$. It is zero under the null and nonzero under the alternative, and becomes larger when the separation between the two distributions increases. Hence, $\\nu_i$ can be viewed as a measure of signal strength.
>
> Second, under the alternative, $\\mu_{g_i}^{(\\mathrm{in})} - \\mu_{g_i}^{(\\mathrm{bet})}$ is the expectation of $\\hat d_i^{(\\mathrm{in})} - \\hat d_i^{(\\mathrm{bet})}$, while $\\Delta_{i,1} - \\Delta_{i,2}$ is the population counterpart of $\\hat{\\Delta}\_{i,1} - \\hat{\\Delta}\_{i,2}$. Thus, $\\nu_i$ can be interpreted as the population analogue of $T_i^2$, so a larger $\\max_{1 \\le i \\le n} \\nu_i$ tends to produce a larger test statistic $T = \\max_{1 \\le i \\le n} T_i^2$ and hence greater power.
>
> Third, we support this intuition with an additional experiment under the same setting as Experiment III. The results show that, as the signal strength increases, both the test statistic $T$ and the empirical power increase accordingly, which is consistent with the above interpretation.
>
> **Table A.3.** Average test statistic and empirical power as functions of the signal strength.
>
> | Type | delta | Statistic | Power |
> |---|---:|---:|---:|
> | Mean | 0.3 | 7.665 | 0.070 |
> | | 0.5 | 9.321 | 0.205 |
> |  | 0.7 | 11.408 | 0.405 |
> |  | 0.9 | 12.596 | 0.630 |
> |  | 1.1 | 12.833 | 0.775 |
> ---

---

> > ### Author Rebuttal · Reviewer_pymn · 2026-04-01
> >
> > My concerns have been adequately addressed.

---

> > > ### Author Response · Authors · 2026-04-02
> > >
> > > Glad to hear all your questions have been answered. Thank you again for the very supportive comments!

---

### Official Review · Reviewer_jYHB · 2026-03-08

**Soundness:** 3
**Presentation:** 3
**Significance:** 3
**Originality:** 3
**Overall Recommendation:** 5
**Confidence:** 3

**Summary:**

This paper expands upon the popular maximum mean discrepancy (MMD) framework and derives a new kernel-based two-sample hypothesis testing of equality of distributions. The authors introduce this test this new test is based on the maximum of differences (MOD) under a kernel-based model. The authors leverage the recent theoretical results from several papers on maximal statistics in high dimensions to develop this test. The authors introduce this toy via a toy example under a location-scale family, under which the MMD to have low power under covariance shift and location shift alternatives. The authors also derived an asymptotic distribution for the test statistic under the null, enabling computation of exact critical values via numerical methods.  The authors also present results under local alternatives of the asymptotic consistency of the test, as well as expand the results into n-variate distributional difference testing. The authors then highlight the test power under several simulated and real data examples stemming from testing distributional differences in the MNIST dataset under different digit distributions.

**Compliance With Llm Reviewing Policy:**

Affirmed.

**Final Justification:**

The authors have adequately addressed my concerns and I am updating my score.

**Key Questions For Authors:**

Does a comparable consistency result exist (as in theorems 3.3/5.2) for the MMD test? An affirmative or negative answer would help contextualize the theoretical contribution of this paper.

Can you discuss the practical difficulty of estimating the n-by-n covariance matrix Sigma (Theorem 3.1) and how estimation error may affect the power of the proposed test in finite samples?

Could you provide a direct ablation comparing MMD and MOD with the same kernels, across a variety of bandwidth choices, without a kernel selection mechanism?

Can the authors provide theoretical justification or a characterization of the distributional shifts where MOD is expected to dominate MMD?

**Limitations:**

Yes

**Strengths And Weaknesses:**

**Soundness**
The theoretical results in this paper appear to be solid and well developed. Theorem 3.1 provides exact asymptotics for the null distribution, and the consistency result against local alternatives (Theorem 3.3). A consistency theorem is provided for the proposed test, enabling principled assessment of power, and the MNIST application provides encouraging empirical support, particularly on the hardest-to-distinguish cases. That said, there is no discussion of the practical difficulty of estimating the n-by-n covariance matrix Sigma, or how estimation error may affect finite-sample power.  Such a comparison against an oracle version where Sigma is known exactly would strengthen the simulation results.

The simulations are also mixed, in some cases MMD outperforms MOD, with no exploration of when MOD should be preferred, and no direct comparison of the two tests using the same kernels. A more thorough ablation across a variety of bandwidth choices, without a kernel selection mechanism, would be informative.

There is also no comparative consistency (as in Theorem 3.3) result for MMD, which limits the strength of the theoretical comparison.

**Presentation**
The location-scale toy example is intuitive, making the paper's motivation easy to follow, and the overall structure is fairly clear. However, several results are cited from external sources (including Theorem 1 from Biggs et al. and results from Chernozhukov et al. ) without being restated. Including these in the appendix, adapted to the paper's own notation, would meaningfully improve clarity.

There are also notation gaps: T_i is used in Section 3 before being defined (though it can be inferred from context), and the structure of the Sigma matrix (line 167) is not made explicit. The median heuristic (Section 6, line 323) is referenced in the experiments but never defined.

Section 2 raises kernel choice dependence for MFD-based tests but drops the thread without development suggesting this paper may address this issue (as it doesn't this may be better held off to the conclusion as a limitation of the method). There is also a duplicate reference entry for Biggs et al. in the appendix bibliography.

**Significance**
The paper addresses a genuine limitation of MMD, and the asymptotic theory and consistency result are meaningful contributions to the two-sample testing literature. The MNIST application demonstrates real-world applicability, particularly on difficult cases. However, the mixed simulation results and the absence of guidance on when to prefer MOD over MMD limit confidence in the practical significance of the contribution.

**Originality**
The connection between theoretical results on maximal statistics and the construction of a new two-sample test is creative.  A clear conceptual contribution rather than a straightforward application of existing techniques. The maximal statistic framing is a well-motivated departure from the aggregated MMD statistic, and the related work appears adequate in situating the contribution relative to prior approaches.

---

> ### Author Rebuttal · Authors · 2026-03-30
>
> **Q1: The practical difficulty of estimating $\Sigma$? Estimation error affects finite-sample power?**
>
> **A1:** Thanks. The covariance matrix $\Sigma$ in our method has an explicit form and only depends on $\Delta_1$ and $\Delta_2$, whose estimators are given on page 4 of the main paper.
>
> We agree that estimating $\Sigma$ may affect finite-sample power. To see this, we conducte a simulation study for Experiment II under the mean-shift alternative and compare the power of the MOD test under the oracle and estimated covariance matrices. Table C.1 in https://anonymous.4open.science/r/paper-rebuttal-materials/README.md  shows that the power difference is very small, suggesting that covariance estimation has only a limited effect on finite-sample performance.
>
> ---
>
> **Q2: MOD vs. MMD across a variety of bandwidth choices**
>
> **A2:** Following your comment, we conducted an additional ablation study. Under Experiment II with $n=250$, we compared the power of MOD and MMD over bandwidths given by the 0.1, 0.3, 0.5, 0.7, and 0.9 quantiles of the pooled pairwise distances, without kernel selection.
>
> Table C.2 at the above link shows that MOD performs better under covariance, location, and distribution shifts, and is only slightly less powerful under mean shift, consistent with the main text. Overall, MOD is effective in detecting non-mean distributional differences.
>
> ---
>
> **Q3: Include several cited results in the appendix**
>
> **A3:** Following your suggestion, we will include these cited results in the appendix, rewritten in notation consistent with the present paper.
>
> ---
>
> **Q4: Notation gaps**
>
> **A4:** Thanks. We will address these notation gaps in the revision by defining $T_i$ when it first appears, making explicit that $\Sigma=(\sigma_{ij})\in\mathbb{R}^{n\times n}$ is the covariance matrix with entries given on page 4, and defining the median heuristic as choosing the Gaussian-kernel bandwidth by the median of pooled pairwise distances.
>
> ---
>
> **Q5: The thread of kernel-choice dependence for MMD-based tests? Duplicate reference entry for Biggs et al.?**
>
> **A5:** Thanks. We agree that the presentation in Section 2 was not sufficiently clear. Our intention was to note the kernel-choice sensitivity of kernel-based tests, review existing remedies for MMD-based procedures, and then motivate MOD-FUSE as an adaptive multi-kernel aggregation method without data splitting. We will clarify this connection in the revision.
>
> We also apologize for the duplicate reference entry in the appendix bibliography and will correct it in the revision.
>
> ---
>
> **Q6: Comparative consistency result for MMD?**
>
> **A6:** To the best of our knowledge, there is no MMD consistency result stated in a form directly comparable to our paper.
>
> Existing MMD consistency results are formulated differently. For example, Gretton et al. (2006) establish consistency against fixed alternatives via the convergence of empirical MMD to its population counterpart, while Gretton et al. (2012) derive a local-alternative result based on RKHS mean-embedding separation. These results are mainly developed for low-dimensional settings.
>
> In high dimensions, Zhu and Shao (2021) show that Gaussian- and Laplacian-kernel MMD permutation tests are consistent only for certain classes of alternatives, while having trivial power for others. Thus, although consistency results for MMD do exist, they are not directly comparable to our theorems.
>
> ---
>
> **Q7: Theoretical justification or a characterization of the distributional shifts where MOD is expected to dominate MMD?**
>
> **A7:** Thanks. A fully general theoretical comparison between MOD and MMD is difficult, since a given distributional shift is hard to translate directly into the corresponding kernel-level discrepancy.
>
> Still, the forms of the two statistics suggest a clear difference in the alternatives they favor. MOD is a max-type statistic and is therefore more sensitive to sparse or localized shifts, whereas MMD is a sum-type statistic based on global aggregation and is thus more naturally suited to dense shifts.
>
> To illustrate this, we conducted a simulation study under a setting similar to Experiment III with $n=300$ and $p=200$, except that only $5\%$ of the observations in $Y$ were drawn from a shifted alternative distribution. Table C.3 at the above link shows that MOD performs better than MMD in this sparse-shift setting. We also plotted the empirical distributions of $T_i$ and $T_i^2$ from one replication; see Figure C.1 at the same link. The plot shows that only a small number of observations produce large local discrepancies.
>
> This interpretation is also consistent with Zhu and Shao (2021), who show that MMD permutation tests can be consistent when cumulative marginal mean or variance differences are sufficiently strong, but may have low or even trivial power when such differences are weak. This also helps explain why MOD tends to outperform MMD under covariance and distribution shifts in our simulations.

---

> > ### Author Rebuttal · Reviewer_jYHB · 2026-04-05
> >
> > Thank you, my concerns have been adequately addressed and I have updated my score.

---

> > > ### Author Response · Authors · 2026-04-06
> > >
> > > Thank you for your positive feedback and for revising the score. We are pleased that our responses have addressed your concerns satisfactorily. We will incorporate all the corresponding revisions and additional results into the final version of the manuscript. We sincerely appreciate your time and constructive comments.

---

### Official Review · Reviewer_r8jq · 2026-03-11

**Soundness:** 3
**Presentation:** 3
**Significance:** 3
**Originality:** 3
**Overall Recommendation:** 4
**Confidence:** 5

**Summary:**

This paper proposes a kernel-based Maximum-of-Difference (MOD) test for two-sample (and $K$-sample) comparison. The method forms a studentized pointwise discrepancy based on within-sample vs.\ between-sample kernel similarities and aggregates via a maximum over observations. The paper motivates MOD by highlighting sign cancellation issues of MMD under covariance/location shifts, provides theoretical guarantees (Gaussian approximation under $H_0$ and consistency under $H_1$), and reports empirical comparisons with several kernel-based baselines including multi-kernel methods.

**Compliance With Llm Reviewing Policy:**

Affirmed.

**Key Questions For Authors:**

## Detailed comments

### 1) Presentation details: notation in Table 1 and theorem numbering

There are a couple of presentation issues that make the paper harder to follow.

First, in Table 1, the symbol $\Delta$ is defined inconsistently across experimental settings: in *Experiment II*, $\Delta$ denotes a *vector-valued* quantity, whereas in *Experiment III*, the same symbol $\Delta$ denotes a *scalar constant*. Reusing the same notation for objects of different types within the same table is confusing.

Second, theorem numbering is inconsistent between the main text and the appendix/proof sections (e.g., “Theorem 1 / Theorem 2” in the appendix versus “Theorem 3.1 / Theorem 3.2” in the main text), which makes cross-referencing unnecessarily difficult.

*Actionable suggestions.*

1. Use distinct symbols for the vector in Experiment II and the scalar in Experiment III (for example, $\boldsymbol{\Delta}$ versus $\delta$ or $\Delta_0$), and explicitly indicate the type or dimension of each parameter in the table.
2. Harmonize theorem and lemma numbering throughout the manuscript and update all cross-references accordingly.

### 2) Dimension dependence in the theory (high-dimensional motivation vs. fixed-$p$ analysis)

From both the introduction and the simulation design, the paper appears to be motivated by high-dimensional settings. However, several rate statements in the proof are expressed only in terms of $n$ (for example, the uniform plug-in error of order $\sqrt{\log n / n}$ in Step 3 of the proof of Theorem 3.1), without explicitly tracking dependence on the ambient dimension $p$ or on possible bandwidth scaling with $p$.

This matters in particular for max-type statistics, where dimension-dependent constants may be amplified when taking maxima. As written, it is therefore unclear whether the theory is intended for a fixed-$p$ regime or whether it is meant to cover growing-dimensional settings.

*Actionable suggestions.*

1. If the intended theory is for fixed $p$, please state this clearly and align the claims and motivation accordingly.
2. If $p = p_n \to \infty$ is allowed, please track dimension-dependent constants explicitly in the concentration bounds (for example, writing the plug-in rate as $K_p \sqrt{\log n / n}$) and state corresponding growth conditions such as $K_p \log n / \sqrt{n} \to 0$.
3. For Gaussian kernels, it would also be helpful to discuss non-degeneracy conditions and possible bandwidth scaling (e.g., $\gamma = \gamma(p)$), to avoid degeneracy of quantities such as $\Delta_1 - \Delta_2$.

### 3) A potentially incorrect independence statement in bounding $I$ (proof of Theorem 3.1)

In the proof of Theorem 3.1, when bounding the term $I$ (and its components $I_1, I_2, I_3$), the argument appears to invoke independence between $H_{ij}(t)$ and $\tilde H^{(ij)}(t)$. I am concerned that this independence may not hold as stated.

Intuitively, $H_{ij}$ depends on $(X_i, X_j)$, while $\tilde H^{(ij)}$—even after removing the $i$- and $j$-indexed parts in the chosen decomposition—may still involve $X_i$ and/or $X_j$ through terms such as $h(X_i, X_k)$ or $h(X_j, X_k)$ for $k \neq i,j$. If so, the two quantities would generally remain dependent.

This point seems central to the correctness of the Gaussian approximation bound, so I believe it should be clarified carefully.

*Actionable suggestions.*

1. Please clarify whether the intended claim is unconditional independence, conditional independence, or independence only with respect to auxiliary Gaussian variables.
2. Please revise the proof accordingly if true independence does not hold under the current decomposition.
3. If needed, consider either restructuring the decomposition to recover independence, or replacing the argument with an appropriate dependence-control technique, such as covariance bounds or a local-dependence Stein argument.

### 4) Clarify the specific role of the *max* aggregation

The paper motivates MOD as a way to address sign cancellation in MMD under covariance-shift or location-shift alternatives. I agree that cancellation can occur for signed sum-type aggregations. However, if the main objective is simply to avoid sign cancellation, then one could also consider nonnegative sum-type alternatives such as

$$
S_2 = \sum_{i=1}^n \big(d_i^{\mathrm{in}} - d_i^{\mathrm{bet}}\big)^2
\quad \text{or} \quad
S_1 = \sum_{i=1}^n \big|d_i^{\mathrm{in}} - d_i^{\mathrm{bet}}\big|,
$$

possibly combined with the same studentization used for $T_i$.

These alternatives would also eliminate sign cancellation. By contrast, the max aggregation corresponds to an $L_\infty$-type scan and usually comes with a different power profile—for example, it may be stronger under sparse or localized alternatives, but potentially weaker under dense and weak alternatives. At present, the paper does not fully explain why the *max* aggregation is methodologically essential, rather than simply one reasonable design choice.

*Actionable suggestions.*

1. Please provide a clearer conceptual explanation of when and why max aggregation is preferable.
2. Ideally, include a small additional experiment or ablation comparing MOD with a studentized nonnegative sum-type alternative, even if only on a subset of the experimental settings.
3. Such a comparison would strengthen the methodological positioning of the paper for an ICML audience.

**Limitations:**

yes

**Strengths And Weaknesses:**

Strengths

1. The central idea is simple and intuitive: use a studentized pointwise discrepancy and scan for the strongest local evidence.

2. The empirical section is fairly comprehensive, with comparisons to a broad set of kernel-based two-sample tests, including multi-kernel methods.

3. The paper attempts to provide theory (null approximation and consistency), which is valuable and helps position the method beyond purely empirical proposals.

Weaknesses

While the approach is promising, I have several concerns that I believe should be addressed for clarity and technical correctness:

1. some presentational details (notation consistency and theorem numbering),

2. the theoretical analysis does not clearly track dimension effects despite an apparently high-dimensional motivation,

3. a potentially incorrect independence claim in the proof of Theorem~3.1, and

4. the paper could more clearly articulate the role of the \emph{max} aggregation relative to other nonnegative aggregations that also avoid sign cancellation.

---

> ### Author Rebuttal · Authors · 2026-03-30
>
> **Q1: Presentation details: notation in Table 1 and theorem numbering**
>
> **A1:**  Thanks. We will revise the notation by using $\Delta \in \mathbb{R}^p$ for the mean shift in Experiment II and $\delta \in \mathbb{R}$ for the scalar signal in Experiment III, and will also unify theorem and lemma numbering.
>
> ---
>
> **Q2: Dimension dependence in the theory**
>
> **A2:**  Thanks for this helpful suggestion. Our method covers both fixed-$p$ and growing-$p$ settings. When $p = p_n \\to \\infty$, the plug-in error bounds in Step 3 of Theorem 1 should be stated more generally as
>
> $$
> \\max_{1 \\le i \\le n}|\\hat\\Delta_{i,1}-\\Delta_1|
> =O_p\\!\\left(K_p\\sqrt{\\frac{\\log n}{n}}\\right),\\qquad
> \\max_{1 \\le i \\le n}|\\hat\\Delta_{i,2}-\\Delta_2|
> =O_p\\!\\left(K_p\\sqrt{\\frac{\\log n}{n}}\\right),
> $$
>
> which yields
>
> $$
> \\left|\\max_{1 \\le i \\le n}Q_i-\\max_{1 \\le i \\le n}T_i\\right|
> =O_p\\!\\left(K_p\\frac{\\log n}{\\sqrt n}\\right),\\qquad
> K_p\\frac{\\log n}{\\sqrt n}\\to 0.
> $$
>
> In the current paper, however, we focus on bounded kernels, such as Gaussian kernels. Under this setting, the concentration bounds hold uniformly, so the proof and rates remain unchanged. We will add a remark in the revision to explain the more general $K_p$-type formulation and why it reduces to the current rates in this setting.
>
> ---
>
> **Q3: Discuss non-degeneracy conditions and possible bandwidth scaling**
>
> **A3:**  Thanks. We agree that bandwidth scaling and the non-degeneracy requirement should be clarified when $p$ grows. In our framework, $\Delta_1-\Delta_2$ must remain bounded away from zero; otherwise, neither the normalization term nor the positive definiteness of $\Sigma$ is guaranteed. If the bandwidth is too large or too small, the Gaussian kernel may become nearly constant or nearly singular, leading to degeneracy. Thus, when $p$ diverges, $\gamma=\gamma(p)$ should scale appropriately with $p$, which also motivates our data-driven choice based on quantiles of pairwise distances.
>
> ---
>
> **Q4: A potentially incorrect independence statement in bounding $I$ (proof of Theorem 3.1)**
>
> **A4:** The independence used in bounding the term $I$ is indeed *unconditional independence* between $\\dot{H}_{ij}(t)$ and $\\tilde{H}^{(ij)}(t)$, and this follows from the two-step leave-one-out method.
>
> Recall that
>
> $$
> \\dot{H}\_{ij}(t)=\\frac{1}{\\sqrt{n}}
> \\left(
> \\frac{1}{\\sqrt{t}}V\_{ij}-\\frac{1}{\\sqrt{1-t}}y\_{ij}
> \\right),
> $$
>
> so it depends only on $(X_i,X_j)$.
>
> To construct $\\tilde{H}^{(ij)}(t)$, we first define
>
> $$
> H^{(ij)}(t)=H(t)-H_i(t)-H_j(t)
> =\\bigl(h_1^{(ij)}(t),\\ldots,h_n^{(ij)}(t)\\bigr)^\\top,
> $$
>
> where
>
> $$
> H(t)=\\sqrt{t}\\,Q+\\sqrt{1-t}\\,Y=\\sum_{k=1}^n H_k(t),
> H_k(t)=\\bigl(H_{1k}(t),\\ldots,H_{nk}(t)\\bigr)^\\top\\in\\mathbb R^n,
> H_{lk}(t)=\\frac{1}{\\sqrt{n}}\\bigl(\\sqrt{t}\\,V_{lk}+\\sqrt{1-t}\\,y_{lk}\\bigr).
> $$
>
> We then define
>
> $$
> \\tilde{H}^{(ij)}(t)=H^{(ij)}(t)-e_i^{(ij)}(t)-e_j^{(ij)}(t),
> $$
>
> where $
> e_r^{(ij)}(t)=\\bigl(0,\\ldots,0,h_r^{(ij)}(t),0,\\ldots,0\\bigr)^\\top\\in\\mathbb R^n.
> $
>
> By construction, $\\tilde{H}^{(ij)}(t)$ removes all terms involving $(X\_i,X\_j)$, whereas $\\dot{H}\_{ij}(t)$ depends only on $(X\_i,X\_j)$. Therefore, $\\dot{H}\_{ij}(t)$ and $\\tilde{H}^{(ij)}(t)$ are independent. Hence, the independence argument used in bounding $I$ is valid.
>
> ---
> **Q5: Clarify the specific role of the max aggregation**
>
> **A5:** Thanks. We agree that nonnegative sum-type statistics also avoid sign cancellation. Following your suggestion, we compared MOD with a studentized sum-type alternative, denoted by SAD: $T_{\mathrm{SAD}}=\sum_{i=1}^n T_i^2.$ Critical values for both MOD and SAD were obtained by permutation.
>
> Table B.1 shows that MOD performs better under mean shift, whereas SAD performs better under covariance shift. To better understand this difference, we further examined the distribution of $T_i^2$ in representative replications; see https://anonymous.4open.science/r/rebuttal-figures-rebuttal/README.md. The pattern is consistent with your comment: under mean shift, the signal is relatively sparse, whereas under covariance shift it is much denser.
>
> However, the sum-type statistic is also harder to analyze theoretically. First, the $T_i$ are strongly dependent, so the limiting law of the sum is much harder to characterize. Second, plug-in errors accumulate across coordinates under summation, making it  difficult to show that the estimation error is asymptotically negligible. By contrast, max aggregation leads to a cleaner and more tractable theory and hence we adopt MOD as the main proposal.
>
> **Table B.1.** Power comparison of MOD and SAD.
>
> | Type | n | MOD | SAD |
> |---|---:|---:|---:|
> | Mean | 100 | 0.38 | 0.20 |
> |  | 150 | 0.56 | 0.26 |
> | | 200 | 0.73 | 0.50 |
> |  | 250 | 0.89 | 0.61 |
> |  | 300 | 0.93 | 0.76 |
> | Covariance | 100 | 0.39 | 0.49 |
> |  | 150 | 0.55 | 0.64 |
> |  | 200 | 0.65 | 0.73 |
> | | 250 | 0.81 | 0.91 |
> |  | 300 | 0.87 | 0.96 |
> ---

---

> > ### Author Rebuttal · Reviewer_r8jq · 2026-04-03
> >
> > No further comments have been provided.

---

> > > ### Author Response · Authors · 2026-04-03
> > >
> > > I am pleased to hear that all questions have been addressed. Thank you once again for your supportive and constructive comments. I will carefully revise the manuscript accordingly.

---

### Decision · Program_Chairs · 2026-04-30

**Decision:**

Accept (regular)

**Comment:**

This paper proposes kernel maximum-of-difference to tackle two-sample problem. The idea is interesting beyond MMD comparison of two distribution where mean difference has been considered. In this paper, the maximum of the difference is used to build test statistics for two-sample testing. Theoretical analysis has been provided and empirical study demonstrates the controlled null and power performances in specified settings. The paper is well-written and would be beneficial the community to appear in ICML.